# Uncertainty Quantification in Retrieval Augmented Question Answering

## Abstract

Retrieval augmented Question Answering (QA) enables QA models to overcome knowledge gaps when answering questions at test time by taking as input the question together with retrieved evidence, that is usually a set of passages. Previous studies show that this approach has numerous benefits such as improving QA performance and reducing hallucinations, without, however, qualifying whether the retrieved passages are indeed useful at answering correctly. In this work, we evaluate existing uncertainty quantification methods and propose an approach that predicts answer correctness based on utility judgements on individual input passages. We train a small neural model that predicts passage utility for a target QA model. We find that simple information theoretic metrics can predict answer correctness up to a certain extent, more expensive sampling based approaches perform better, while our lightweight approach can efficiently approximate or improve upon sampling-based approaches.

## 1 Introduction

Retrieval augmented Question Answering (QA) enables QA models to overcome knowledge gaps when answering user questions at test time by giving them access to input evidence, i.e., a set of passages, retrieved for the user questions (Lewis et al., 2020; Guu et al., 2020; Izacard et al., 2024). Recent work exploits the language understanding and generation abilities of Large Language Models (LLMs; (Brown et al., 2020; Ouyang et al., 2024)) and makes use of external retrievers to find the input evidence (Chen et al., 2017; Izacard & Grave, 2021a). That is, the retrieved evidence is given to the LLM-based QA model as input context in tandem with the question; the QA model will read this evidence and formulate an answer. For instance, in Figure 1, for the user question *Who sings Does He Love Me with Reba?*, the QA model is provided with a set of evidence passages together with the question; and correctly formulates the answer *Linda Davis*.

Such retrieval augmented QA architectures have proven beneficial enabling access to external knowledge (Izacard et al., 2024), increasing the performance on tail knowledge (Mallen et al., 2023), reducing hallucinations in model answers, and even improving model calibration (Jiang et al., 2021). However, there are various ways in which a retrieval augmented QA approach can go wrong at production time. The set of passages obtained using retrieval methods is far from perfect (Sciavolino et al., 2021; Yoran et al., 2024; Kasai et al., 2024) containing irrelevant or misleading evidence, the model might be under-trained to read certain passages and reason over these and the question (Izacard et al., 2024; Liu et al., 2024b), or the question can simply be ambiguous or unanswerable (Kasai et al., 2024). In these cases where the QA system lacks the knowledge to formulate an answer (i.e., it is uncertain about what the answer is), we want it to refrain from answering rather than providing an erroneous answer. Thus, predicting answer uncertainty is key.

Approaches to answer uncertainty prediction can be grouped in two main categories, sampling- and LLM-based methods. Sampling-based methods to QA uncertainty detection rely on the output discrepancies among multiple predictors on the same input (Gal & Ghahramani, 2016; Lakshminarayanan et al., 2017); i.e., this variance in outputs indicates that the model is uncertain. Concretely, these methods sample via temperature scaling (Guo et al., 2017) and then measure diversity on the set of sampled answers (Kuhn et al., 2023; Chen & Mueller, 2024). These approaches are expensive to run for in-production QA systems and the quality of the semantic similarity will degrade on long

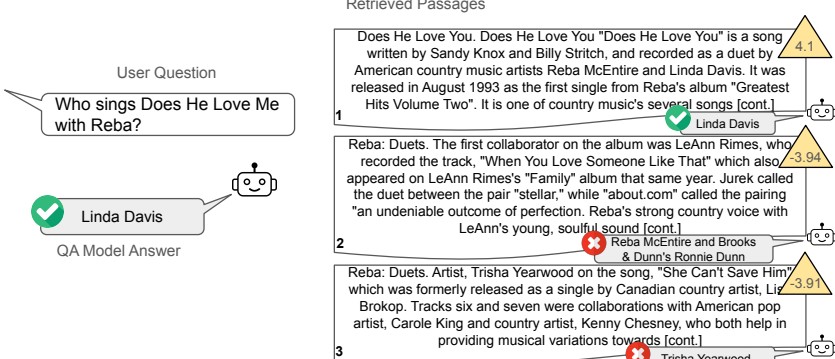

Figure 1: Example of user question from the Natural Questions dataset with the set of three top retrieved passages with Contriever (Izacard et al., 2022) (the other two passages below the rank are less relevant and not shown in the figure); the gold answer is *Linda Davis*. The target QA model GEMMA2-9B correctly answers the question when provided with the top five passages. Below each passage, it is shown the answer generated by the QA model when only prompted with that passage and the question. The QA model correctly answers when prompted with the first passage and produces an incorrect answer when prompted with each of the other ones. The yellow triangles on the top right of the passages are the predicted utility scores by our utility ranker. Higher values indicate more useful passages and our model correctly identifies that the top passage is better.

answers (Zhang et al., 2024).[1] LLM-based methods explore to what extent language models are able to correctly express uncertainty about their own predictions (Kadavath et al., 2022; Lin et al., 2022; Tian et al., 2023; Zhou et al., 2023). These look into whether the model's confidence in its outputs coincides with their correctness (i.e., calibration), methods to fix calibration, and ways to elicit from the model a verbal expression of that confidence (i.e., linguistic calibration). Findings about model calibration are diverse and model dependent, fixing relies on approximations for the case of black-box models and fine-tuning what could be infeasible in practice given current LLMs' sizes. None of these answer uncertainty detection approaches has been applied in the context of retrieval augmented QA, most of them are applied on closed-book QA tasks where the answer is predicted based on the question and the models' encoded knowledge.

In this work, we propose a **secondary model that makes predictions at individual retrieved passage level that are useful to estimate answer uncertainty of retrieval augmented QA models**. We hypothesize that the type of retrieved passages and questions, the relation between them and their implicit interaction with the QA model's own knowledge are indicators of answer correctness. If the passages are informative and priming the QA model towards appropriate knowledge, we expect the QA model to produce a correct answer. In contrast, if the passages are not informative or misleading and the posed question is out of the QA model's knowledge, we expect it to generate an erroneous answer (i.e., either factually incorrect or completely made up content). We operationalise this as retrieved passage *utility*. Given a question, a passage is *useful*, if a QA model can correctly answer the question based on it. We train a small neural model to predict passage utility which we refer to as *utility ranker*. We train the utility ranker on utility judgements generated by the target QA model. We borrow ideas from direct uncertainty quantification approaches (Van Amersfoort et al., 2020; Lahlou et al., 2023) but we do not decompose uncertainty or outline shifts in the input distribution.

**We show that individual passage utilities are good predictors of retrieval augmented QA accuracy. This means that it is possible to train an answer uncertainty predictor independently from the choice of number of retrieved passages used to prompt the target QA model.** Because retrieved passages are scored individually, our approach is independent of the *number* of retrieved passages chosen for the target QA model. We evaluate our approach on short-form question answering tasks. Figure 1 shows an example of input question, set of retrieved passages, and correct answer

---

[1]By expensive we mean both latency as well as cost as a long prompts might need to be processed and QA systems may rely on paid proprietary language models.

from the Natural Questions dataset (Kwiatkowski et al., 2019). Results on six QA datasets show that our approach performs on par with existing sampling-based uncertainty quantification approaches while being more efficient at test time. It requires a small model pass over the set of input passages and the question (see inference cost comparison in Appendix C.1). Surprisingly, in more complex reasoning questions (SQuAD) and adversarial QA settings (e.g., rare entities or unanswerable questions) our approach surpasses existing uncertainty quantification methods. Moreover, we show that the utility scores predicted by the Utility Ranker can be used to re-rank retrieved passages obtained via the external retrieval system to improve QA accuracy (Liu et al., 2024b).

## 2 RELATED WORK

**Uncertainty Quantification for Question Answering**   Several methods have been proposed to predict answer uncertainty in QA; however, none of them has analysed uncertainty in retrieval augmented QA models. Many existing approaches rely on capturing output variation as the expression of model uncertainty (Kuhn et al., 2023; Farquhar et al., 2024; Chen & Mueller, 2024). On a sample of model outputs, Kuhn et al. (2023) propose to first cluster answers with similar meaning via natural language inference before computing entropy. Chen & Mueller (2024) propose an approach for *black-box* models, they also compute similarities in the set of answers but associate them with a model self-judgement of confidence. These approaches are expensive to run at inference time for a production QA system, they require several inference steps plus the similarity computations. In addition, as the length of the answers increases, measuring similarity becomes more complex (Zhang et al., 2024). Hou et al. (2024) propose a decomposition of predictive uncertainty and focus on quantifying aleatoric uncertainty (i.e., uncertainty in the data) caused by ambiguous questions. This approach is orthogonal to ours.

**Judging the Utility of Retrieved Passages**   Previous work has analysed the set of retrieved passages (Yu et al., 2023; Asai et al., 2024; Wang et al., 2024; Xu et al., 2024; Yoran et al., 2024) following the observation that passages can be irrelevant or misleading making the QA model prone to producing incorrect answers. Asai et al. (2024) make use of an external critic model to judge whether a question requires retrieval (or not), whether the retrieved passages are relevant to formulate the answer, and whether the final response elaborated by the QA model is useful. While they analyse retrieved passage *relevance*, this decision is taken by an external extreme-scale critic (e.g., GPT-4) and used to fine-tune the QA model. In contrast, we do not fine-tune the target QA model but rather we elicit utility judgements from it to train a secondary model to predict passage utility. Other work creates auxiliary tasks around retrieved passages enforcing the QA model to reason on them; e.g., by taking notes about each passage (Yu et al., 2023) or generating passage summaries (Xu et al., 2024). These methods also use extreme-scale LLMs to generate training data to fine-tune the retrieval augmented QA model. Park et al. (2024) select specific in-context examples to improve the LLM's reasoning on the input passages, their focus is on detecting input passages with conflicting content (e.g., different dates for a given event). These approaches aim at improving QA performance while our primary goal is modelling QA uncertainty.

**Improving Retriever via Reader Performance**   Previous work with pre-trained language models has focused on jointly training the retriever and reader modules end-to-end (Lee et al., 2019; Lewis et al., 2020; Izacard & Grave, 2021b). That is, the performance of the question answering model is propagated also to the retriever. This joint training scheme can be very expensive for current (extreme-scale) LLMs. Our approach can be seen as an intermediate module between the QA model (reader) and the external retriever. It would be interesting to explore our utility ranker to provide feedback (e.g., to label data) to fine-tuning the retriever. In recent work, Salemi & Zamani (2024) evaluate the performance of information retrieval systems via retrieval augmented QA performance. Interestingly, they show that external judgements (e.g., query-document relevance labels) of passage utility correlate poorly with retrieval augmented QA performance.

**Learning to Predict Confidence**   Some approaches train a specific model to predict a confidence score (Dong et al., 2018; Kamath et al., 2020; Mielke et al., 2022). For semantic parsing, Dong et al. (2018) train a confidence predictor based on a set of uncertainty features from the input and the model. Mielke et al. (2022) also train a calibrator that, given the user question and model generated answer, predicts a confidence score. In our approach, we simple aggregate predicted individual

passage utilities but it would also be possible to train a confidence module that takes utilities with other features into account (e.g., output sequence probability), and predicts a confidence score.

# 3 MODELLING ANSWER UNCERTAINTY

Formally, we define retrieval augmented QA as follows. Given question $x$ and set of retrieved passages $R = \{p_1, p_2, \cdots, p_{|R|}\}$ obtained with retriever $\mathcal{R}$, a LLM-based QA model $\mathcal{M}$ is prompted to generate answer $y_{\mathcal{M}}$ to question $x$ token-by-token as $y_{\mathcal{M}} = \arg\max_{y_{\mathcal{M}}} \prod_{t=1}^{|y_{\mathcal{M}}|} p_{\mathcal{M}}(y_t | y_{1..t-1}, x, R)$. We want to estimate the uncertainty or error of $\mathcal{M}$ on generating $y_{\mathcal{M}}$ given $x$ and $R$; i.e., we want an estimator $\{x, R\} \mapsto \mathbf{u}_{\mathcal{M}}(\{x, R\})$ of $\mathcal{M}$'s answer uncertainty. In our approach, the answer uncertainty predictor $\mathbf{u}_{\mathcal{M}}$ is based on individual passage utilities. Our hypothesis is that individual passage utilities of retrieved passages in $R$ are indicators of the QA model uncertainty when generating $y_{\mathcal{M}}$ when prompted with $R$. For instance, in Figure 1, given that the first passage in the set has a high utility score, this indicates that the QA model is likely to be confident when providing the answer *Linda Davis*. Thus, we want a passage utility estimator $\{x, p\} \mapsto \upsilon_{\mathcal{M}}(\{x, p\})$ of every $p \in R$. In what follows, we define passage utility and how to estimate and predict it. Next, we discuss a simple answer uncertainty estimator $\mathbf{u}_{\mathcal{M}}$ based on $\upsilon_{\mathcal{M}}$.

## 3.1 PASSAGE UTILITY RANKING

**Passage Utility** Intuitively, a passage $p$ retrieved for question $x$ is useful for a QA model $\mathcal{M}$, if $\mathcal{M}$ can correctly answer $x$ when prompted with $p$. In addition, $\mathcal{M}$'s reliance on passage $p$ to formulate the answer may vary. That is, the QA model may formulate a correct answer even though $p$ does not provide the answer itself; instead, $p$ positively primes $\mathcal{M}$ to use its memorised knowledge. The utility of the first passage, in Figure 1, is high as the QA model generates a correct answer when prompted with it and the fact that *Linda Davis* sings together with *Reba McEntire* can be derived from it. The second and third passages, although related to the topic of the question, are not useful. The QA model is potentially uncertain about how to answer the question and the passages do not help; the model incorrectly answers when prompted with each of them. Thus, the utility of the second and third passages is low.

Concretely, we estimate the utility of passage $p$ for QA model $\mathcal{M}$ to answer question $x$ by combining two measures. These are *accuracy*, denoted as $a(y_{\mathcal{M}})$, whether the generated answer $y_{\mathcal{M}}$ is correct, and *entailment*, denoted as $e(y_{\mathcal{M}})$, how much does passage $p$ supports the generated answer $y_{\mathcal{M}}$. Accuracy is computed by a critic model $\mathcal{A}$ and entailment by a Natural Language Inference (NLI) classifier model $\mathcal{E}$. We define the combined passage utility as $\upsilon_{\mathcal{M}} = (a(y_{\mathcal{M}}) + e(y_{\mathcal{M}}))/2$ that takes values in the closed interval $[0, 1]$ given that $a$ takes values in the set $\{0, 1\}$ and $e$ in the closed interval $[0, 1]$.

**Utility Ranker** We train a small neural model to predict passage utility scores, $\{x, p\} \mapsto \upsilon_{\mathcal{M}}(\{x, p\})$. We use observed answer accuracy and entailment by QA model $\mathcal{M}$ on a training set $D = \{(x, p)\}$ to train the utility predictor. That is, we run the QA model $\mathcal{M}$ on examples from $D$ and compute passage utilities to form a training set for our utility predictor $D_{\mathcal{M}} = \{(x, p, \upsilon_{\mathcal{M}})\}$.

For recall purposes, retrieval augmented QA generally retrieves more than one input passage for each question $x$, i.e., $|R| > 1$. To generate training data for the passage utility predictor, we retrieve $|R|$ passages per question in order for to cover passages with different usefulness. From the set of passages $R$ for question $x$, we derive training instances $\{(x, p_i, \upsilon_{\mathcal{M}i}) | p_i \in R\}$. We exploit this to train the passage utility predictor with a contrastive learning scheme. That is, if $p_i$ and $p_j$ are passages in $R$ and $p_i$ is more useful than passage $p_j$ to answer question $x$, the predicted utility score $\upsilon_{\mathcal{M}i}$ should be higher by a margin $m$ than the predicted score $\upsilon_{\mathcal{M}j}$ for $p_j$ (i.e., $p_i$ should be ranked higher than $p_j$). We train the utility predictor with the following pair-wise ranking objective:

$$\mathcal{L}_{rank} = \sum_{((x,p_i),(x,p_j)) \in R \times R, i \neq j} \max(0, -z(\upsilon_{\mathcal{M}i} - \upsilon_{\mathcal{M}j}) + m)), \tag{1}$$

where $z$ controls the gold order between $p_i$ and $p_j$ (e.g., if $z = 1$, $p_i$ has higher utility, conversely $z = -1$ indicates the opposite ordering) and $m$ is a hyper-parameter. The passage utility predictor is trained with a Siamese neural network. Its architecture is constituted by a BERT (Devlin et al.,

2019) based encoder followed by a pooling and two MLP layers stacked on top of BERT outputs (Fang et al., 2024). The output layer computes the utility score as $v_{\mathcal{M}i} = W_o h^L + b_o$ where $h^L$ is the vector representation for $(x, p_i)$ from the last hidden layer (the L-th layer) of the network. At inference time, we compute a single utility score for each passage. We provide implementation and training details in Section 4.

To enforce the signal on accuracy prediction and to regularise the range of utility values learned by the ranking scheme, we combine the ranking objective in Equation 1 with the following Binary Cross Entropy (BCE) objective (Sculley, 2010):

$$\mathcal{L}_{BCE} = \sum_{(x,p) \in \{(x,p_i),(x,p_j)\}} a_{\mathcal{M}} \times (\log(p(x,p)) + (1 - a_{\mathcal{M}}) \times \log(1 - p(x,p)), \qquad (2)$$

where $p(x,p) = \text{sigmoid}(v_{\mathcal{M}})$ and $a_{\mathcal{M}}$ is the target accuracy label taking values in the set $\{0, 1\}$. We train the utility predictor with the following combined objective:

$$\mathcal{L} = \mathcal{L}_{rank} + \lambda \mathcal{L}_{BCE}, \qquad (3)$$

where $\lambda$ is a hyper-parameter. Both the ranking and BCE objectives are compatible with gold annotations that could be obtained via human intervention in an interactive and active learning learning setting. That is, it would be feasible to elicit from human judges (e.g., moderators of the QA system) answer accuracy labels (e.g., *correct/incorrect*) and level of passage support for the generated answer (e.g., *best* or *worse*) (Simpson et al., 2020; Fang et al., 2024). Note that the Utility Ranker could also be trained with different variants of this objective that also exhibit competitive performance. We report in Appendix D.1 a study on the ablation of the different components of the training objective.

The passage utility predictor is related to the direct error prediction approach in (Lahlou et al., 2023). Lahlou et al. (2023) train a secondary model to estimate target model loss; instead, we train the passage utility predictor with sequence level metrics, i.e., accuracy and entailment, which indirectly measure error. This choice is best suited for our task for various reasons. First, in the context of text generation and its possibly diverse (e.g., paraphrases) but correct set of possible generated answers (Kuhn et al., 2023), predicting loss against a unique single paraphrase would result in a too narrow estimation. Our choice is also adequate for proprietary LLMs where it is not possible to create training data with model losses. Finally, our approach is suited for collecting data from user feedback for active model adaptation (Simpson et al., 2020; Fang et al., 2024). In the image domain, van Amersfoort et al. (2020) map inputs to feature representations and take the distance between new inputs and their closest cluster centroids as a measure of uncertainty. In retrieval augmented QA with LLMs, text passages, and questions, it is less clear what the boundary between seen and unseen texts or topics is. Because our Utility Ranker is trained on a target dataset it could be exploited to detect out-of-domain instances for a target application. It would be interesting to pursue future work on using our Utility Ranker as a content controller for the target LLM-based QA model.

Some approaches to answer uncertainty prediction that train a secondary model are in (Kamath et al., 2020; Zhang et al., 2021). However, none of them is applied to retrieval augmented QA; but instead to Reading Comprehension (RC), i.e., the task of generating an answer based on a single positive (i.e., supposed to contain the answer) context document. There are two major differences with our work. One is that in their scenario, all input documents are useful while in ours the utility of retrieved passages is varied. The second one is that we show that individual passage utilities are good predictors of retrieval augmented QA with a set of retrieved passages.

## 3.2 Answer Uncertainty Estimation for Retrieval Augmented QA

For retrieval augmented QA, we want an estimator $\{x, R\} \mapsto \mathbf{u}_{\mathcal{M}}(\{x, R\})$ of the answer uncertainty of a target QA model $\mathcal{M}$ when generating answer $y_{\mathcal{M}}$ from a prompt with set of passages $R$ and question $x$. We propose the direct estimation of $\mathbf{u}_{\mathcal{M}}$ from individual passage utilities predicted for passages in $R$. The intuition is that, the highest the utility in one (or more) passages in $R$ the less uncertain $\mathcal{M}$ will be when generating answer $y_{\mathcal{M}}$. Concretely, we take the maximum utility score that is given to passages in $R$ as an estimate of answer uncertainty $\mathbf{u}_{\mathcal{M}}$, i.e.,

$$\mathbf{u}_{\mathcal{M}}(\{x, R\}) = \max(v_{\mathcal{M}}(\{x, p\}) \,|\, p \in R). \qquad (4)$$

Note that other more complex estimators $\{x, R\} \mapsto \mathbf{u}_{\mathcal{M}}(\{x, R\})$ could be learned by training, for instance, a regression model on individual passage utilities in addition to other features of the target model $\mathcal{M}$ such as probability of the generated answer $y_{\mathcal{M}}$ (Dong et al., 2018).

## 4 EXPERIMENTAL SETUP

**Accuracy Evaluation** A precise metric for measuring accuracy is key when evaluating the quality of uncertainty estimation. Token overlap metrics are far from being precise and can over- or under-estimate accuracy, e.g., Acc yields a higher score for the pair of gold and generated answers (*a politician*, *not a politician*) than for the pair (*a politician*, *a congressperson*). Thus, our main metric to evaluate QA model performance and as the accuracy evaluator $\mathcal{A}$ to create data to train the passage utility predictor, is based on a LLM judgement of accuracy proposed by Sun et al. (2024) (**AccLM**). A critic LLM is prompted with the gold and generated answer and asked to judge whether they are the equivalent. In a sample of 840 generated answers human and LLM-based judgment of correctness agreed 98% of the time (Sun et al., 2024). We use the prompt as proposed in (Sun et al., 2024), we include it in Appendix B for completeness. We use Qwen2-72B-Instruct (Yang et al., 2024) to obtain accuracy judgments. For compatibility with previous work and as a lower bound, in Appendix D.2, we report QA model performance with token overlap accuracy (**Acc**) defined as whether the gold answer is contained in the generated answer (Mallen et al., 2023; Asai et al., 2024).

**Utility Ranker Implementation Details** To create the training set $D_{\mathcal{M}}$ to train the Utility Ranker, we consider the first top five retrieved passages for each question, i.e., $|R| = 5$. Note that this is a hyper-parameter and other values would also be possible, e.g., with larger sizes of $|R|$ further training data would be available. We use the target QA model $\mathcal{M}$ to generate answers $y_{\mathcal{M}}$ for each of the five passages $p$ in $R$ (i.e., $\mathcal{M}$ is prompted with passage $p$ and question $x$). We then ge utility scores using the LLM-based accuracy judge $\mathcal{A}$ as described above and an ALBERT-xlarge Lan et al. (2020) model optimized on MNLI (Williams et al., 2018) and VitaminC (Schuster et al., 2021) as our entailment judge $\mathcal{E}$.

**Comparison Approaches and Baselines** We choose the stronger methods from previous work (Fadeeva et al., 2023) to compare our approach with.

*Information Based.* We compare against the stronger information based uncertainty quantification approaches reported in previous work Fadeeva et al. (2023). These are based on predictive probabilities; recall that the predictive distribution under QA model $\mathcal{M}$ prompted with question $x$ and set of passages $R$ is $P(y_{\mathcal{M}}|x, R, \mathcal{M}) = \prod_{t=1}^{|y_{\mathcal{M}}|} p_{\mathcal{M}}(y_t|y_{1..t-1}, x, R)$ for a target QA model $\mathcal{M}$.

Maximum Sequence Probability (MSP) based uncertainty estimation is based on the probability of the most likely answer and computed as $\mathrm{MSP}(y_{\mathcal{M}} \,|\, x, R, \mathcal{M}) = 1 - P(y_{\mathcal{M}}|x, R, \mathcal{M})$. The other uncertainty estimation approach is the negative mean Point-wise Mutual Information (PMI) Takayama & Arase (2019); i.e., it compares the probability of generating answer $y_{\mathcal{M}}$ given the prompt with question $x$ and passages $R$ w.r.t the probability given by $\mathcal{M}$ to $y_{\mathcal{M}}$ without context. Intuitively, the higher the PMI the more certain on generating $y_{\mathcal{M}}$. PMI is computed as $PMI(y_{\mathcal{M}}, x, R; \mathcal{M}) \frac{1}{|y_{\mathcal{M}}|} \sum_{t=1}^{|y_{\mathcal{M}}|} \log \frac{p_{\mathcal{M}}(y_t|y_{1..t-1})}{p_{\mathcal{M}}(y_t|y_{1..t-1}, x, R)}$. The other two methods are based on entropy. We compare with Regular Entropy (RE), i.e., the entropy on the predictive distribution computed at sequence level $\mathbb{E}[-\log P(y_{\mathcal{M}}|x, R, \mathcal{M})]$ with $\mathbb{E}$ computed on sequences $y_{\mathcal{M}}$ sampled from $P(y_{\mathcal{M}} \,|\, x, R, \mathcal{M})$. In practice, this is approximated via Monte-Carlo integration, i.e., sampling $N$ random answers from $P(y_{\mathcal{M}} \,|\, x, R, \mathcal{M})$. Thus, Regular Entropy is computed as $-\frac{1}{N} \sum_{n=1}^{N} \log \tilde{P}(y_{\mathcal{M}}^{(n)} \,|\, x, R, \mathcal{M})$, where $\tilde{P}(y_{\mathcal{M}}^{(n)} \,|\, x, R, \mathcal{M})$ is the length normalised version of $P(y_{\mathcal{M}}^{(n)}|x, R, \mathcal{M})$.

*Answer Variation.* Kuhn et al. (2023) propose a variant of regular entropy, named Semantic Entropy (SE), that accounts for uncertainty in the surface form of the generated answers rather than on meaning. Concretely, Semantic Entropy clusters the set of $N$ samples into $M$, $M \leq N$, clusters with the same meaning via bidirectional entailment. Then computes the average answers' probability within each cluster, $SE(x, \mathcal{M}) = -\sum_{m=1}^{M} \hat{P}_m(x, \mathcal{M}) \log \hat{P}_m(x, \mathcal{M})$ where $\hat{P}_m(x, \mathcal{M}) = \frac{\sum_{y_{\mathcal{M}} \in C_m} P(y_{\mathcal{M}} \,|\, x, R, \mathcal{M})}{\sum_{m=1}^{M} \sum_{y_{\mathcal{M}} \in C_m} P(y_{\mathcal{M}} \,|\, x, R, \mathcal{M})}$.

*Reflexive.* We compare with p(true) proposed by Kadavath et al. (2022). This approach uses the same target QA model (LLM) evaluate whether the answers it produces are correct. It is prompted with the question and a set of candidate answers, i.e., the most likely answer plus a sample of size $N$ answers,

and instructed to respond whether the most likely answer is true or false (i.e., correct/incorrect). The score produced by this approach is the probability of the model $\mathcal{M}$ generating the token True. p(true) needs several in-context examples to work well, so we fit as many examples as can be in the context.

*Baselines.* The sets of passages in $R$ are originally ranked by the IR system, so each passage in $R$ has a retriever score which can be seen as baseline passage utility. We thus take the Retriever Score as a baseline. Despite the QA models are instructed to produce a short answer, these often generate longer answers. The length of the answer could be a feature indicating that the model is uncertain about the answer. Thus, we estimate answer uncertainty from the Answer Length (Ans.Len) as the number of words in the answer.

Following previous work (Farquhar et al., 2024), we take $N = 10$ samples and use multinomial sampling to generate samples. That is, we set the sampling temperature to 1, with nucleus sampling ($P = 0.9$) (Holtzman et al., 2020) and top$-K$ sampling ($K = 50$) (Fan et al., 2018), and use a different random seed to draw each sample. Most likely answers are generated with greedy sampling at temperature equal to 0. We use the implementation provided by Farquhar et al. (2024) to compute RE, SE, CA, and p(true). We report inference cost of each approach in Appendix C.1.

**QA Models** Our target retrieval augmented QA models $\mathcal{M}$ are based on the following instruction fine-tuned LLMs. To assess the performance of the Utility Ranker for QA models that potentially exhibit different answer uncertainty, we consider different families of similar size. These are Llama-3.1-8B-Instruct (AI@Meta, 2024), Mistral-7B-Instruct-v0.3 Jiang et al. (2023), and Gemma2-9B-it Riviere et al. (2024). For all QA models, we use a simple prompt including the retrieved passages and the question in the context, the prompt is shown in Table 6 of the Appendix. We use vLLM for inference (Kwon et al., 2023). Following previous work on retrieval augmented QA, we use Contriever Izacard et al. (2022) as our external retriever (Asai et al., 2024) and the target QA models are prompted with $|R| = 5$ passages Yu et al. (2023); Asai et al. (2024); Xu et al. (2024).

**Datasets** We evaluate our answering uncertainty prediction approach on short-form answer generation tasks. Concretely, we evaluate on the Natural Questions Kwiatkowski et al. (2019), TriviaQA Joshi et al. (2017), WebQuestions Berant et al. (2013), and SQuAD (Rajpurkar et al., 2016) datasets. We follow the training/validation/test splits in prior work Lee et al. (2019); Min et al. (2019); Karpukhin et al. (2020). To test the generalisation robustness of our approach we carry out additional experiments on PopQA Mallen et al. (2023), a dataset with questions about rare entities, and RefuNQ Liu et al. (2024a), a dataset with unanswerable questions about non-existing entities. Statistics about our datasets are given in the Appendix in Table 5.

**Evaluation of the Quality of Uncertainty Estimation** To assess the quality of answer uncertainty prediction, we follow Farquhar et al. (2024) and report the Area Under the Receiver Operator Curve on detecting answer uncertainty, i.e., incorrect answers, (**AUROC**) and the area under the rejection accuracy curve (**AURAC**). AURAC summarises the accuracy of QA models when answer uncertainty is used to refuse to answer questions. It summarises accuracy at different percentages of rejection. Instruction fine-tuned models are known to refuse to answer questions, i.e., they produce answers such as *This information is not available in the text*. In some cases, the refusal response will be adequate (e.g., no input passage contains the information to answer) but in many cases QA models may refuse when they should have provided an answer Adlakha et al. (2024); Liu et al. (2024a). Thus, to simplify the assessment of answer correctness, we did not explicitly instruct the QA models to abstain and treat occurring refusal answers as cases of uncertainty where the QA model is expressing the uncertainty in the answer (Farquhar et al., 2024). We report the percentage of refusal answers for each QA model and QA task on development sets in Appendix D.2.

## 5 RESULTS

### 5.1 UNCERTAINTY QUANTIFICATION

Answer uncertainty estimation results for the three QA models (GEMMA2-9B, LLAMA-3.1-8B, and MISTRAL-7B-V0.3) are shown in Table 1 (results on the development set are included in Appendix D). In terms of predicting answer uncertainty (i.e., model incorrect answers), column AUROC in Table 1, simple metrics based on models' probabilities such as MSP perform better for some

Table 1: Answer uncertainty estimation for QA models GEMMA2-9B, LLAMA-3.1-8B, and MISTRAL-7B-V0.3 on NaturalQuestions, TriviaQA, WebQuestions, and SQuAD (evaluation with in-distribution test data for the Utility Ranker). We report AUROC and AURAC.

| | NaturalQuestions | | TriviaQA | | WebQuestions | | SQuAD | |
|---|---|---|---|---|---|---|---|---|
| | AUROC | AURAC | AUROC | AURAC | AUROC | AURAC | AUROC | AURAC |
| GEMMA2-9B | | | | | | | | |
| MSP | 0.69 | 0.67 | 0.68 | 0.83 | 0.63 | 0.66 | 0.65 | 0.63 |
| PMI | 0.51 | 0.58 | 0.53 | 0.78 | 0.45 | 0.58 | 0.50 | 0.55 |
| p(true) | 0.72 | 0.70 | 0.78 | 0.86 | **0.74** | **0.74** | 0.67 | 0.66 |
| Regular Entropy | 0.69 | 0.68 | 0.65 | 0.82 | 0.63 | 0.67 | 0.65 | 0.62 |
| Cluster Assignment | 0.67 | 0.66 | 0.69 | 0.83 | 0.60 | 0.65 | 0.66 | 0.63 |
| Semantic Entropy | 0.68 | 0.67 | 0.68 | 0.83 | 0.60 | 0.65 | 0.66 | 0.63 |
| Ans.Len | 0.65 | 0.65 | 0.59 | 0.80 | 0.62 | 0.66 | 0.61 | 0.60 |
| Retriever Score | 0.60 | 0.65 | 0.68 | 0.84 | 0.53 | 0.62 | 0.61 | 0.62 |
| Utility Ranker | **0.76** | **0.72** | **0.81** | **0.88** | 0.72 | 0.71 | **0.81** | **0.74** |
| LLAMA-3.1-8B | | | | | | | | |
| MSP | 0.71 | 0.69 | 0.83 | **0.88** | 0.71 | 0.74 | 0.77 | 0.69 |
| PMI | 0.56 | 0.60 | 0.57 | 0.78 | 0.51 | 0.65 | 0.61 | 0.59 |
| p(true) | **0.79** | **0.74** | **0.84** | 0.87 | **0.76** | 0.76 | 0.65 | 0.61 |
| Regular Entropy | 0.72 | 0.69 | 0.83 | **0.88** | 0.72 | 0.74 | 0.78 | 0.69 |
| Semantic Entropy | 0.69 | 0.67 | 0.81 | 0.86 | 0.68 | 0.73 | 0.75 | 0.68 |
| Ans.Len | 0.59 | 0.61 | 0.60 | 0.78 | 0.61 | 0.68 | 0.57 | 0.55 |
| Retriever Score | 0.58 | 0.62 | 0.64 | 0.81 | 0.50 | 0.63 | 0.65 | 0.61 |
| Utility Ranker | 0.73 | 0.70 | 0.78 | 0.86 | 0.76 | **0.78** | **0.84** | **0.73** |
| MISTRAL-7B-V0.3 | | | | | | | | |
| MSP | 0.68 | 0.63 | 0.73 | 0.87 | 0.65 | 0.68 | 0.71 | 0.65 |
| PMI | 0.53 | 0.59 | 0.55 | 0.79 | 0.50 | 0.62 | 0.58 | 0.60 |
| p(true) | 0.72 | 0.67 | **0.84** | **0.88** | 0.72 | 0.70 | 0.69 | 0.63 |
| Regular Entropy | 0.60 | 0.60 | 0.71 | 0.85 | 0.61 | 0.68 | 0.66 | 0.62 |
| Semantic Entropy | 0.67 | 0.63 | 0.78 | **0.88** | 0.69 | 0.68 | 0.71 | 0.66 |
| Ans.Len | 0.68 | 0.63 | 0.67 | 0.84 | 0.64 | 0.69 | 0.66 | 0.63 |
| Retriever Score | 0.59 | 0.60 | 0.67 | 0.82 | 0.53 | 0.65 | 0.64 | 0.62 |
| Utility Ranker | **0.76** | **0.68** | 0.79 | 0.86 | **0.76** | **0.71** | **0.80** | **0.68** |

models. It exhibits high performance for LLAMA-3.1-8B while lower performance for GEMMA2-9B and MISTRAL-7B-V0.3. Sampling-based approaches (meaning diversity and reflexive), can better identify model uncertainty but at the cost of running inference several times to have a good size sample for the estimation. Our Utility Ranker has similar or better performance with a single inference step on each input passage. We speculate that clustering approaches can suffer in phrase or sentence level correct answers where these contain different levels of details Zhang et al. (2024); thus, not being clustered together wrongly suggesting variation.

On improving question-answering accuracy, AURAC column in Table 1, with the exception of TriviaQA, all uncertainty prediction approaches outperform the information theoretic approaches (i.e., MSP, PMI). The Utility Ranker performs on par or better than the more expensive sampling based approaches. To have a clearer picture of baseline retrieval augmented QA accuracy w.r.t. accuracy when the uncertainty estimation is used to decide whether to abstain nor not, we show in Figure 2 the accuracy of the model at different thresholds for answer rejection. That is, we report when the QA model chooses to answer only the 80% or 90% of the most confident cases as well as when always answers. Retrieval augmented QA accuracy per model and dataset on the full test and development sets is included in Appendix D. Across all datasets, the Utility Ranker performs on par with of better than more expensive uncertainty estimation approaches. The easiest QA task is TriviaQA where QA models show very good performance and information theoretic methods work on par with more complex ones. On the most difficult task, SQuAD, the utility ranker outperforms all other methods both at 20% and 10% of rejected answers.

## 5.2 ROBUSTNESS AND GENERALISATION OF UNCERTAINTY ESTIMATION

We assess the robustness and generalisation of our Utility Ranker on test cases that are different from those examples seen during training, i.e., Out-Of-Distribution (OOD). These examples encompass real cases that a QA model will face at test time such as different type of questions, e.g., longer and more complex. We also study extreme adversarial cases such as questions about tail knowledge for both retrievers and LLMs (PopQA) and unanswerable questions (RefuNQ).

**Distribution Shift** Table 2 shows the performance of the Utility Ranker when evaluated in OOD data. The first column indicates the training data and the first row the evaluation data. Results in the

Figure 2: Average QA model performance on test sets with $|R| = 5$. We show model based accuracy (AccLM) at different percentages of rejecting to answer (i.e., when choosing to respond on 80%, 90%, and all the cases) given uncertainty estimations by the different approaches.

Table 2: Performance of GEMMA2-9B's Utility Ranker on distribution shift. That is, trained on one dataset and evaluated zero-hot on another one. We report all combinations of train and test data. The first column indicates train data while the first row test data.

| | NaturalQuestions | | TriviaQA | | WebQuestions | | SQuAD | |
| --- | --- | --- | --- | --- | --- | --- | --- | --- |
| | AUROC | AURAC | AUROC | AURAC | AUROC | AURAC | AUROC | AURAC |
| NaturalQuestions | **0.76** | **0.72** | 0.72 | 0.86 | 0.65 | 0.67 | 0.72 | 0.68 |
| TriviaQA | 0.64 | 0.67 | **0.81** | **0.88** | 0.63 | 0.68 | 0.71 | 0.68 |
| WebQuestions | 0.60 | 0.64 | 0.72 | 0.86 | **0.72** | **0.71** | 0.58 | 0.59 |
| SQuAD | 0.65 | 0.67 | 0.77 | 0.87 | 0.61 | 0.65 | **0.81** | **0.74** |

diagonal correspond to the Utility Ranker trained and evaluated in the same data distribution; the off-diagonal cells to the Utility Ranker evaluated zero-shot in a different dataset. As expected, the Utility Ranker variants evaluated on a different dataset show a decrease in performance. However, for some training data the decrease is small providing a competitive prediction. That is, NaturalQuestions and SQuAD provide the best training data, what agrees with previous experiments in reading comprehension settings (Chen et al. (2021) choose NaturalQuestions to train the base model, Kamath et al. (2020); Zhang et al. (2021) SQuAD). The Utility Ranker variants trained on WebQuestions (smallest training set) and TriviaQA (the easiest task) have the worst generalisation performance. Note that we focus on zero-shot to assess bare transfer performance; however, it would make sense to train the model with few examples of the OOD data (Kamath et al., 2020; Zhang et al., 2021).

**Adversarial Questions** Table 3 reports results for GEMMA2-9B's Utility Ranker trained on NaturalQuestions and evaluated zero-shot to predict answer uncertainty for retrieval augmented QA with $|R| = 5$ on PopQA and RefuNQ. These datasets are made of adversarial cases so we report AUROC (predicting incorrect answers) and AURAC (summary of different rejection thresholds). For RefuNQ as questions have (un)answerable gold annotations, we further report AUROC scores for the Unanswerable questions (51% of the tested cases) and all incorrect answers together (67%) of the test cases. The Utility Ranker (NQ) outperforms other methods to detect answer uncertainty across datasets and improve QA accuracy by refusing to answer questions across the board. We attribute this to the fact that, either due to knowledge about tail entities (PopQA) or unanswerable questions about nonexistent concepts (RefuNQ), the quality of the retrieved passages suffers. Thus, our approach will assign lower utility to these and thus successfully predict answer uncertainty. This is confirmed by the surprisingly high AUROC score achieved by the Retriever Score baseline on Re-

Table 3: Answer uncertainty estimation for GEMMA2-9B on adversarial QA tasks (PopQA and RefuNQ). Its Utility Ranker is trained on Natural Questions.

| | PopQA | | RefuNQ | | |
| --- | --- | --- | --- | --- | --- |
| | AUROC | AURAC | AUROC | | AURAC |
| | | | All | Unanswerable | |
| MSP | 0.66 | 0.58 | 0.66 | 0.63 | 0.39 |
| PMI | 0.51 | 0.50 | 0.54 | 0.53 | 0.35 |
| p(true) | 0.71 | **0.62** | 0.73 | 0.65 | 0.45 |
| Regular Entropy | 0.66 | 0.58 | 0.66 | 0.61 | 0.39 |
| Semantic Entropy | 0.69 | 0.59 | 0.68 | 0.60 | 0.41 |
| Ans.Len | 0.62 | 0.55 | 0.65 | 0.66 | 0.38 |
| Retriever Score | 0.63 | 0.58 | 0.76 | **0.80** | 0.47 |
| Utility Ranker (NQ) | **0.72** | **0.62** | **0.82** | 0.71 | **0.51** |

Table 4: Retrieval augmented QA performance with three passages $|R| = 3$ is the version with the top three retrieved passages from Contriever and $|R^{UR}| = 3$ is the version with top three re-ranked passages out of ten originally retrieved. We report model based (AccLM) accuracy.

| | NaturalQuestions | TriviaQA | WebQuestions | SQuAD |
| --- | --- | --- | --- | --- |
| $|R|$ = top 3 ranked by external retriever | 0.58 | 0.77 | 0.63 | 0.53 |
| $|R|$ = top 3 re-ranked by Utility Ranker | 0.62 | 0.79 | 0.65 | 0.60 |
| $|R|$ = all 10 passages | **0.64** | **0.80** | **0.66** | **0.62** |

fuNQ's Unanswarable questions. In this particular type of questions, the retrieval system indeed struggles to retrieve relevant passages. Note that Retriever Score behaves otherwise in the rest of the QA tasks where it shows lower performance. Interestingly, information based methods, MSP and PPL, perform worse in these adversarial QA tasks than in the in-distribution test cases (Section 5.1). This shows that in these cases QA models produce incorrect answers with high confidence.

## 5.3 IMPROVING QA PERFORMANCE

We also assess the quality of the passage utility scores to identify informative passages via end task QA performance. We compare the original ranking by the external retrieval system with the ranking established by the utility scores by taking the top 3 passages out of 10 passages ordered by the external retriever and re-ranked by the Utility Ranker. We then run the QA models with a budget of $|R| = 3$ input passages. We also run the QA model with the all the 10 passages, i.e., with $|R| = 10$. Results for GEMMA2-9B QA model are shown in Table 4. The QA model with the top 3 passages re-ranked by the Utility Ranker improves 4 points on NaturalQuestions and 7 points on SQuAD over the QA model variant that takes the top 3 ranked by the external retriever. This suggest that passages considered relevant for user questions by the external retriever do not coincide with what is useful for the target QA model. The QA model variant with the top 3 passages re-ranked by the Utility Ranker performs very close, i.e., difference of 1 or 2 points across all datasets, to the QA model variant with the 10 passages given as context. The utility scores effectively identify informative passages and the QA model achieves comparable performance with a much shorter prompt.

## 6 CONCLUSIONS

In this work we present an approach to answer uncertainty prediction for retrieval augmented QA models. Importantly, this approach relies on single passage utilities. This approach is based on a small neural model that is trained on a target QA model judgements of retrieved passage usefulness. We show that this approach is competitive or better than existing strong error prediction approaches while being light-weight. Our experiments also show that our approach is particularly better in cases of extreme QA model answer uncertainty like rare entities and unanswerable questions. Future work would explore the approach in the context of log-form generation tasks, e.g., query focused-generation. It would also be interesting to explore to what extent the Utility Ranker model could be used in active learning scenarios.

### 6.1 ETHICS STATEMENT

Our work does not involve human subjects. We use QA datasets that are publicly available and widely used by the research community.

## 6.2 REPRODUCIBILITY STATEMENT

We build up on existing base code Farquhar et al. (2024); Fang et al. (2024) and we will make available all code and data together with the docker images for reproducibility.

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

| Dataset | Train | Dev | Test |
|---|---|---|---|
| Natural Questions | 79,168 | 8,757 | 3,610 |
| TriviaQA | 78,785 | 8,837 | 11,313 |
| WebQuestions | 2,474 | 361 | 2,032 |
| SQuAD | 78,713 | 8,886 | 10,570 |
| PopQA | 11267 | - | 3000 |
| RefuNQ | - | - | 4439 |

Table 5: Dataset statistics, number of instances per train/dev/test sets. Note that we sample a smaller test set for PopQA in our experiments.

---

Knowledge:

[1] [passage]
[2] [passage]
...
[N] [passage]

Answer the following question with a very short phrase.

Question: [question]

---

Table 6: Minimal prompt selected as user turn for the QA models.

pp. 1958–1970, Online, August 2021. Association for Computational Linguistics. doi: 10.18653/v1/2021.findings-acl.172. URL https://aclanthology.org/2021.findings-acl.172.

Kaitlyn Zhou, Dan Jurafsky, and Tatsunori Hashimoto. Navigating the grey area: How expressions of uncertainty and overconfidence affect language models. In Houda Bouamor, Juan Pino, and Kalika Bali (eds.), *Proceedings of the 2023 Conference on Empirical Methods in Natural Language Processing*, pp. 5506–5524, Singapore, December 2023. Association for Computational Linguistics. doi: 10.18653/v1/2023.emnlp-main.335. URL https://aclanthology.org/2023.emnlp-main.335.

## A  DATASETS

Table 5 shows statistics about the QA datasets we use in our experiments.

## B  MODEL PROMPTS

The prompt we use for our QA models is shown in Table 6. Table 7 illustrate the prompts used for our LLM based accuracy and p(true) baseline.

## C  COMPARISON AND BASELINE UNCERTAINTY ESTIMATION METHODS

### C.1  TEST TIME COST OF UNCERTAINTY ESTIMATION METHODS

Table 8 shows the cost of executing uncertainty estimation for a user question $x$ in terms of model inference calls required. Simple information theoretic methods require a single call (PPL, MSP) or two (PMI) calls to the QA model with the full prompt ($N$ retrieved passages and user question $x$); similarly the Ans.Len baseline. However, approaches that estimate uncertainty based on diversity (Regular Entropy, Cluster Assignment, Semantic Entropy, and p(true)) require generating $N$ answers, i.e., $N$ inference calls with the full prompt. In addition, Cluster Assignment and Semantic Entropy require the computation of the answers clusters, so additional calls to an entailment model

are required to compare the set of sampled answers. p(true) requires one additional LLM call to elicit a True/False answer but with a very long prompt including in-context examples and candidate answers. In contrast, our approach requires $|R|$ utility predictions with a BERT-size model.

# D  ADDITIONAL RESULTS

## D.1  DIFFERENT COMPONENTS OF THE TRAINING OBJECTIVE

Table 9 shows results on the ablation of the Utility Ranker training objective (Section 3.1, Equation 3). When trained only with the ranking loss ($\mathcal{L}_{rank}$), in average it achieves better performance when the training signal combines accuracy ($a$) with entailment ($e$), i.e., the training ranking is given by $(e+a)/2$. When trained in combination with the full objective ($\mathcal{L}_{rank} + \mathcal{L}_{BCE}$) the ranker shows an increase of 10 AUROC points. Highlighting the benefit of training the Utility Ranker to predicting QA accuracy for input passages. Interestingly, when we drop the ranking loss (i.e., last line of Table 9) there is a drop in performance. On one hand, the ranking loss enables the comparison of pairs of passages and thus the number of training instances is higher. On the other hand, the entailment -based ranking signal might help the final model to learn features useful for more accurate passage utility prediction.

## D.2  UNCERTAINTY ESTIMATION RESULTS

Table 10 and 11 shows retrieval augmented QA performance on the development set for the target QA models. Table 12 shows performance of uncertainty quantification approaches. We report AURAC and AUROC as well as the percentage out of incorrect cases where the QA models produce an answer acknowledging the lack of knowledge to answer.

We report the following additional uncertainty estimation methods. Perplexity (PPL) computed as $PPL(y_\mathcal{M}, x, R; \mathcal{M}) = \exp\left\{-\frac{1}{|y_\mathcal{M}|}\sum_{t=1}^{|y_\mathcal{M}|} P_\mathcal{M}(y_t|y_{1..t-1}, x, R)\right\}$, i.e., based on the average negative log-likelihood of the generated tokens. Cluster Assignment (CA) is the variant of SE without answers' probabilities where $\hat{P}_m(x, \mathcal{M})$ is approximated from the number of answers in the cluster. CA values are very close to SE values.

You need to check whether the prediction of a question-answering system to a question is correct. You should make the judgment based on a list of ground truth answers provided to you. Your response should be "correct" if the prediction is correct or "incorrect" if the prediction is wrong.

Question: Who authored The Taming of the Shrew (published in 2002)?
Ground truth: ["William Shakespeare", "Roma Gill"]
Prediction: W Shakespeare
Correctness: correct

Question: Who authored The Taming of the Shrew (published in 2002)?
Ground truth: ["William Shakespeare", "Roma Gill"]
Prediction: Roma Gill and W Shakespeare
Correctness: correct

Question: Who authored The Taming of the Shrew (published in 2002)?
Ground truth: ["William Shakespeare", "Roma Gill"]"
Prediction: Roma Shakespeare
Correctness: incorrect

Question: What country is Maharashtra Metro Rail Corporation Limited located in?
Ground truth: ["India"]
Prediction: Maharashtra
Correctness: incorrect

Question: What's the job of Song Kang-ho in Parasite (2019)?
Ground truth: ["actor"]
Prediction: He plays the role of Kim Ki-taek, the patriarch of the Kim family.
Correctness: correct

Question: Which era did Michael Oakeshott belong to?
Ground truth: ["20th-century philosophy"]
Prediction: 20th century."
Correctness: correct

Question: Edward Tise (known for Full Metal Jacket (1987)) is in what department?
Ground truth: ["sound department"]
Prediction: 2nd Infantry Division, United States Army
Correctness: incorrect

Question: What wine region is Finger Lakes AVA a part of?
Ground truth: ["New York wine"]
Prediction: Finger Lakes AVA
Correctness: incorrect

Question: [question]
Ground truth: [answers]
Prediction: [output]
Correctness:

Table 7: Prompt for accuracy evaluation.

| | Nb./Type of Inference Call at Test Time |
|---|---|
| PPL | $1\,$LLM-G |
| MSP | $1\,$LLM-G |
| PMI | $2\,$LLM-G |
| p(true) | $(N+1)\,$LLM-G $+ 1\,$LLM-E |
| Regular Entropy | $(N+1)\,$LLM-G |
| Cluster Assignment | $(N+1)\,$LLM-G $+ N(N-1)/2\,$LLM-E |
| Semantic Entropy | $(N+1)\,$LLM-G $+ N(N-1)/2\,$LLM-E |
| Ans.Len | $1\,$LLM-G |
| Retriever Score | $0\,$LLM-G |
| Utility Ranker | $|R|\,$Bert-F |

Table 8: Number and type of inference call required to estimate answer uncertainty for a given user question $x$. LLM-G means inference with the retrieval augmented QA model, i.e., a forward pass with the prompt including the set of $R$ retrieved passages and the question to generate an answer. LLM-E is inference with an evaluation model, e.g., a forward pass to ask a LLM for correctness in p(true) or a forward pass with an entailment model in the Semantic Entropy method. Bert-F is an inference call to predict passage utility for a passage $p$ in $R$ and user question $x$.

Table 9: Uncertainty Estimation by the Utility Ranker trained with variants of the training objective. We report AUROC and AURAC for the Utility Ranker for the three target QA models (GEMMA2-9B, LLAMA3.1-8B, and MISTRAL-7B-V0.3) on Natural Questions development data.

| | GEMMA2-9B | | LLAMA3.1-8B | | MISTRAL-7B-V0.3 | |
|---|---|---|---|---|---|---|
| | AUROC | AURAC | AUROC | AURAC | AUROC | AURAC |
| $\mathcal{L}_{rank}, (e+a)/2 + \mathcal{L}_{BCE}$ | 0.77 | 0.76 | 0.77 | 0.76 | 0.79 | 0.76 |
| $\mathcal{L}_{rank}, (e+a)/2$ | 0.67 | 0.70 | 0.66 | 0.70 | 0.69 | 0.70 |
| $\mathcal{L}_{rank}, (a)$ | 0.62 | 0.67 | 0.64 | 0.68 | 0.67 | 0.69 |
| $\mathcal{L}_{rank}, (e)$ | 0.67 | 0.70 | 0.64 | 0.68 | 0.64 | 0.67 |
| $\mathcal{L}_{BCE}$ | 0.76 | 0.74 | 0.75 | 0.74 | 0.77 | 0.74 |

Table 10: Target QA models performance on test sets with $|R| = 5$. Model based accuracy AccLM (column header ALM) is accuracy computed by Qwen2-72B-Instruct.

| | NaturalQuestions | | TriviaQA | | WebQuestions | | SQuAD | | PopQA | | RefuNQ | |
|---|---|---|---|---|---|---|---|---|---|---|---|---|
| | Acc | ALM | Acc | ALM | Acc | ALM | Acc | ALM | Acc | ALM | Acc | ALM |
| GEMMA2.9B | 0.46 | 0.61 | 0.73 | 0.78 | 0.40 | 0.64 | 0.41 | 0.58 | 0.49 | 0.51 | 0.26 | 0.35 |
| LLAMA-3.1-8B | 0.47 | 0.60 | 0.71 | 0.77 | 0.44 | 0.66 | 0.41 | 0.56 | 0.48 | 0.49 | 0.27 | 0.37 |
| MISTRAL-7B-V0.3 | 0.47 | 0.58 | 0.71 | 0.75 | 0.47 | 0.66 | 0.40 | 0.57 | 0.52 | 0.49 | 0.27 | 0.35 |

Table 11: Target QA models performance on the development sets with $|R| = 5$. (Acc) is rule based accuracy as used in previous work, (AccLM) is accuracy computed by Qwen2-72B-Instruct.

| | Natural Questions | | TriviaQA | | WebQuestions | | SQuAD | |
|---|---|---|---|---|---|---|---|---|
| | Acc | AccLM | Acc | AccLM | Acc | AccLM | Acc | AccLM |
| GEMMA2.9B | 0.45 | 0.62 | 0.73 | 0.79 | 0.45 | 0.67 | 0.37 | 0.58 |
| LLAMA-3.1-8B | 0.46 | 0.60 | 0.71 | 0.77 | 0.52 | 0.68 | 0.38 | 0.58 |
| MISTRAL-7B-V0.3 | 0.46 | 0.60 | 0.71 | 0.76 | 0.53 | 0.69 | 0.36 | 0.57 |

Table 12: Answer uncertainty estimation for QA models GEMMA2-9B, LLAMA-3.1-8B, and MISTRAL-7B-V0.3 on NaturalQuestions, TriviaQA, WebQuestions, and SQuAD development sets. We report AUROC and AURAC. Refusal % is the percentage out of the total incorrect answers where the model acknowledges uncertainty by expressing its lack of knowledge in the generated answer.

| | Natural Questions | | TriviaQA | | WebQuestions | | SQuAD | |
|---|---|---|---|---|---|---|---|---|
| | AUROC | AURAC | AUROC | AURAC | AUROC | AURAC | AUROC | AURAC |
| | GEMMA2-9B | | | | | | | |
| PPL | 0.67 | 0.69 | 0.61 | 0.80 | 0.63 | 0.70 | 0.65 | 0.66 |
| MSP | 0.69 | 0.70 | 0.66 | 0.81 | 0.64 | 0.70 | 0.66 | 0.66 |
| PMI | 0.49 | 0.59 | 0.42 | 0.71 | 0.49 | 0.63 | 0.46 | 0.55 |
| p(true) | 0.73 | 0.73 | 0.76 | 0.85 | 0.73 | 0.75 | 0.70 | 0.69 |
| Regular Entropy | 0.70 | 0.69 | 0.66 | 0.81 | 0.65 | 0.70 | 0.68 | 0.68 |
| Cluster Assignment | 0.70 | 0.70 | 0.67 | 0.81 | 0.65 | 0.70 | 0.65 | 0.66 |
| Semantic Entropy | 0.71 | 0.71 | 0.65 | 0.80 | 0.65 | 0.71 | 0.65 | 0.66 |
| Ans.Len | 0.63 | 0.66 | 0.62 | 0.79 | 0.61 | 0.69 | 0.60 | 0.64 |
| Retriever Score | 0.59 | 0.65 | 0.62 | 0.80 | 0.50 | 0.62 | 0.67 | 0.68 |
| Utility Ranker | **0.75** | **0.74** | **0.79** | **0.86** | **0.74** | **0.77** | **0.82** | **0.77** |
| Refusal % | 5% | | 5% | | 0.7% | | 3% | |
| | LLAMA3.1-8B | | | | | | | |
| PPL | 0.75 | 0.75 | 0.80 | 0.85 | 0.68 | 0.73 | 0.71 | 0.70 |
| MSP | 0.79 | 0.77 | 0.83 | 0.86 | 0.69 | 0.73 | 0.72 | 0.70 |
| PMI | 0.61 | 0.68 | 0.56 | 0.75 | 0.55 | 0.67 | 0.55 | 0.60 |
| p(true) | 0.79 | 0.77 | **0.89** | **0.88** | 0.72 | 0.75 | 0.69 | 0.69 |
| Regular Entropy | **0.81** | **0.78** | 0.82 | 0.86 | 0.69 | 0.74 | 0.75 | 0.72 |
| Cluster Assignment | 0.77 | 0.75 | 0.82 | 0.85 | 0.72 | 0.75 | 0.75 | 0.72 |
| Semantic Entropy | 0.76 | 0.75 | 0.84 | 0.86 | 0.71 | 0.75 | 0.76 | 0.73 |
| Ans.Len | 0.63 | 0.67 | 0.66 | 0.79 | 0.61 | 0.69 | 0.56 | 0.60 |
| Retriever Score | 0.57 | 0.65 | 0.62 | 0.78 | 0.49 | 0.64 | 0.67 | 0.67 |
| Utility Ranker | 0.79 | 0.77 | 0.81 | 0.85 | **0.77** | **0.79** | **0.83** | **0.76** |
| Refusal % | 2% | | 1% | | 0.7% | | 2% | |
| | MISTRAL-7B-V0.3 | | | | | | | |
| PPL | 0.65 | 0.69 | 0.65 | 0.80 | 0.62 | 0.70 | 0.66 | 0.65 |
| MSP | 0.70 | 0.71 | 0.74 | 0.82 | 0.67 | 0.73 | 0.72 | 0.68 |
| PMI | 0.49 | 0.60 | 0.57 | 0.76 | 0.56 | 0.68 | 0.54 | 0.58 |
| p(true) | 0.73 | 0.71 | **0.80** | **0.85** | 0.69 | 0.75 | 0.70 | 0.67 |
| Regular Entropy | 0.65 | 0.69 | 0.66 | 0.80 | 0.63 | 0.71 | 0.70 | 0.68 |
| Cluster Assignment | 0.71 | 0.72 | 0.76 | 0.82 | 0.71 | 0.75 | 0.75 | 0.69 |
| Semantic Entropy | 0.72 | 0.72 | 0.77 | 0.83 | 0.71 | 0.74 | 0.75 | 0.70 |
| Ans.Len | 0.65 | 0.68 | 0.69 | 0.80 | 0.64 | 0.72 | 0.66 | 0.64 |
| Retriever Score | 0.59 | 0.65 | 0.61 | 0.77 | 0.58 | 0.69 | 0.64 | 0.63 |
| Utility Ranker | **0.76** | **0.74** | 0.77 | 0.84 | **0.73** | **0.77** | **0.80** | **0.72** |
| Refusal % | 1% | | 0.25% | | 3% | | 0.5% | |

