# OpenReview forum: "Uncertainty Quantification in Retrieval Augmented Question Answering"
_ICLR.cc/2025/Conference — Submitted to ICLR 2025_

### Official Review · Reviewer_575d · 2024-11-03

**Soundness:** 2
**Presentation:** 2
**Contribution:** 2
**Rating:** 5
**Confidence:** 5

**Summary:**

This paper evaluates existing uncertainty quantification approaches that are used to quantify whether the retrieved passages at test time are useful to answer the question correctly. This paper also propose a neural-model based utility ranker that predicts answer correctness based on utility judgements on individual input passage, which boost the accuracy by 4% for NQ, and 2% for WebQ. The utility ranker trumps some uncertainty detection method for some datasets for the Gemma model. However, more analysis and explanations could be done.

**Strengths:**

- Previous work for QA error detection are either expensive to run for in-production QA systems, or rely on model’s internal confidence, and are for closed-book QA, so they are not applicable to retrieval augmented setup. The proposed utility ranker,
- The utility ranker differs from Asai et al. (2024) because Asai et al. (2024) uses external critic model to judge, while the proposed method is target QA model based.
- The baseline methods are thoroughly tested in both table 2, table 3 and table 4.
- After applying utility ranker to filter out irrelevant passages, the accuracy and accuracy LM increases for both model on NQ and WebQ.

**Weaknesses:**

- Section 3 overcomplicates the method, and some of the math definitions are confusing instead of explaining the details. If I understand it correctly, the usefulness of a passage $p$ is defined on whether the model can correctly answer the question with the passage. The utility score is defined as the mean of accuracy and entailment score, where both scores are binary, so the only possible value for $u$ is 0, 0.5, or 1. Then it is combined with a binary cross entropy objective, and train a Siamese network that uses DistilRoBERTa to encode text pairs, and then use the ALBERT-xlarge model, trained on MNLI and VitaminC, is used to determine the entailment.
- Result analysis could be done for section 5.3. Although both Acc and AccLM is improved by utility ranker, some explanations is appreciated. Why is the analysis only done for NQ and WebQ? Both Accuracy and AccuracyLM increases the same amount, is it a coincidence or one metric is enough.

**Questions:**

- Is the $m$ in equation 1 a hyper-parameter?
- In equation 2, the summation is over $u_i$ and $u_j$, but they didn’t show up in the equation. You also mention $p(y) = \sigmoid(u)$, but is it $u_i$ or $u_j$?
- The citation format needs fixing, not limited to:
    - Line 208: (PMI; Takayama & Arase, 2019), as well as the citation for p(true) on line 211 needs fixing.
    - Line 218: Holtzman et al. 2020.
    - Line 224, Gemma2-9B-Instruct (Riviere et al., 2024), and line 227 for contriever.
    - Line 232: You can use \citep for multiple citations, and use comma to separate each citation.
- Missing citation: Top-k sampling is from Fan et al. (2018).
- Why do you select |R| = 3 for table 5 rather than |R| = 5?
- Is there analysis about when and what |R| people should use? Does the effect enhance or decrease when |R| change? Is the method still relevant if there are more than X number of passages?
- It seems like utility ranker works better on Gemma rather than Llama, if experiment could be run on other models to confirm that utility ranker does work for most models, that would be wonderful.

## Reference

Fan, Angela, Mike Lewis, and Yann Dauphin. "Hierarchical neural story generation." *arXiv preprint arXiv:1805.04833*(2018).

---

> ### Author Response · Authors · 2024-11-20
>
> ## Weaknesses
>
> - Indeed, the usefulness of a passage $p$ is defined on whether the model can correctly answer the question with the passage. We would like to clarify the computation of the passage utility score. It is effectively defined as the mean of the accuracy and entailment score, with the accuracy being a binary value, hence, a value from the set {0, 1}, whereas the entailment score corresponds to the posterior probability of the entailment class, hence, a value in the interval [0, 1]. We further take the average because initially we wanted to have a final utility score also in the interval [0, 1]. However, it would also be possible to take the summation of both, in the end, the Utility Ranker is trained with a ranking objective. The only requirement is that the utility score should be able to order passages according to their usefulness. For instance, passages that lead to an accurate response and have high entailment should have higher utility that those that have midle entailment and lead to an inaccurate response.
>
> - 2) Additional results and explanation for Section 5.3
>
> We added results for TriviaQA and SQuAD which show a similar trend (Table below).
>
> |------|NaturalQuestions| |TriviaQA| |WebQuestions|  |SQuAD| |
> |------|------|------|------|------|------|------|------|------|
> |      | Acc | AccLM | Acc | AccLM | Acc | AccLM | Acc | AccLM |
> | $size-of(R)=3$ | 0.43 | 0.58 | 0.71 | 0.77 | 0.38 | 0.63 | 0.38 | 0.53 |
> | $size-of(R^{urank})=3$ | **0.47** | **0.62** | **0.73** | **0.79** | **0.40** | **0.65** | **0.44** | **0.60** |
>
> It is a coincidence about the same increase. If incorrect answers with $|R|=3$ happen to turn to be correct with $|R^{urank}|=3$ and the answer string matches the gold answers then will also add up the same increments as the LLM one. For this experiment (data and models) maybe one metric is enough to show the gain.
>
>
>
> ## Questions
>
> - 1) and 2)
>
> $m$ in Equation 1 is a hyper-parameter set to 0.1 in all our experiments.
>
> In Equation 2, we compute the BCE loss for each pair of passages $p_i$ and $p_j$ in $R$. $\sigmoid(u)$ means applied to $u_i$ and $u_j$. We will rewrite this en the new version of the pdf for clarity.
>
> - 2) and 3) references will be fixed in the new version of the pdf.
>
> - 4) and 5) Choice of $R$.
>
> We selected a small $|R|$ (i.e., $|R|=3$) because we want to evaluate the re-ranking by the utility ranker (and show the differences w.r.t. the original retriever ranking) on few top passages. If the number of top passages increases and is large then it is less visible the effect of re-ranking. The goal of doing re-ranking, is to improve performance with the smallest number of input passages possible. For in production QA systems, the smaller the context the better both for cost and latency purposes.
>
> There is no (to the best of our knowledge) a study about what values of $|R|$ should be used. However, we chose $|R|=5$ for our main experiments based on the following facts. First, we follow most of existing work on retrieval augmented QA that uses $|R|=5$ (e.g., [1], [2], [3]). Second, LLMs may exhibit poor behaviour when reading long contexts, thus ([4]), the smaller and more precise the set of passages the better. Finally, as mentioned in the previous paragraph and inspired by authors' knowledge of real scenario practices in industry products, the smaller the number of passages the better.
>
> [1] Chain-of-note: Enhancing robustness in retrieval-augmented language model
> [2] RECOMP: IMPROVING RETRIEVAL-AUGMENTED LMS WITH COMPRESSION AND SELECTIVE AUGMENTATION
> [3] Self-RAG: Learning to retrieve, generate, and critique through self-reflection
> [4] Lost in the Middle: How Language Models Use Long Contexts
>
> - 6) We add results for a model of different family but similar size, i.e., Mistral-7B-Instruct-v0.3. See response to Reviewer trrB.

---

> > ### Comment · Reviewer_575d · 2024-11-25
> > **Response to the authors**
> >
> > Thank you for your response!
> >
> > Re: W1 -- Thank you for the clarification for the possible values of the utility score. I hope the math can be more polished in the next version.
> > Re: W2 -- Thank you for these additional results, and the increases for both models are indeed similar.
> > Re: |R| choice -- Thank you for the pointers and the explanations. My questions are not relevant to the choice of |R| = 5, since I was merely curious of the reason of choosing |R|=5 for almost all main experiments yet choosing |R| = 3 for this specific experiment. However, I believe that the comparison is still not fair to the baseline, as the utility ranker has the information of the top 10 paragraphs, yet the baseline doesn't.
> >
> > An additional question after reading the other reviews and responses: I understand why there is no accuracy/AccLM results on RefuQA, but why are there no accuracy/AccLM results on PopQA?

---

> > ### Author Response · Authors · 2024-11-27
> >
> > - Re-ranking experiment.
> >
> > We would like to clarify that the main contribution of our work is to show that it is possible to predict answer uncertainty for retrieval augmented QA from individual input passage utility.
> >
> > The passage utility score establishes an order among  passages retrieved for a given question. We show that the order is meaningful as it can improve retrieval augmented QA accuracy. There is information gain, that is the main goal of re-ranking, the Utility Ranker does a good job on keeping the most important passages within the top 3. We clarify the description of the experiment in Section 5.3 and add an additional baseline where the QA model generates with the 10 passages given as input (on the updated pdf). This new results show that accuracy with the top 3 re-ranked by the Utility Ranker is better than the top 3 ranker by the original retrieval system; and that accuracy with the top 3 re-ranked passages is close to the accuracy (1/2 points) with the 10 passages in the context.
> >
> > -  AURAC and Acc/AccLM metrics.
> >
> > Thank you for your suggestion on the presentation of the results. We agree that reporting AccLM at X% (and base AccLM) in addition to AURAC makes results clearer. We have eliminated Table 1 (kept it in the Appendix for reference) and we include the AccLM at different levels of rejection. In the new version of the pdf that we upload this can be found in Figure 1.

---

> ### Author Response · Authors · 2024-11-22
> **Response Submitted**
>
> Dear Reviewer, thank you for your useful comments and recommendations. We have submitted a response addressing all your points and would like to have your acknowledgement/feedback and continue discussions. Kind regards, Authors.

---

> ### Author Response · Authors · 2024-11-25
>
> Thank you for the acknowledgement and feedback on our response.
>
> - RE:W1 we will upload a new version of the pdf.
>
> - $|R|=3 experiment $.
>
> We want to clarify that the primary goal of this experiment is to show that the ranking by the Utility Ranker is better that the original ranking by the retrieval system. As we do not have gold passage order annotations to directly compare, for instance with Hit@N metric, we compare this via end-task performance (i.e., retrieval augmented QA accuracy).
>
> - We reported accuracy for PopQA and RefuNQ in the main paper, L301. We will include these results in  Table 1 to make them more visible.

---

> > ### Comment · Reviewer_575d · 2024-11-25
> > **Response to the authors**
> >
> > Thank you for the response! I will retain my score for now:
> >
> > Re: |R|=3 experiment -- thank you for the response. Judging from the task performance and your updated results, indeed the $|R^{urank}|=3$ performs better than $|R|=3$. However, I don't think this sufficiently answers my question of whether this comparison is fair to the baseline or not. Since the utility ranker indeed has intake more information (10 passages by LM) than the baseline (3 passages by LM), it is not clear that the better performance of utility ranker is due to this additional information gain, or due to utility ranker.
> >
> > Re: accuracy for PopQA and RefuNQ: thank you for the pointer! However, I still think the overall presentation of this paper needs significant improvements, for example, the baseline comparisons are only done for AUROC and AURAC. I strongly advise to report the Acc/AccLM score for the baseline too, as mentioned in your response, end-task performance could reflect the reranker's effectiveness, and I believe it might be more of people's interest than the threshold metrics, and more beneficial for selling the idea of your UtilityRanker. Furthermore, table 1 and table 9 seem just for demonstrating the model performance (without utility ranker) and justifying the choice of using AccLM as the metric, and therefore a bit out-of-place.

---

> ### Comment · Reviewer_575d · 2024-11-27
> **Response to the authors**
>
> Thank you for updating the pdf and answering my questions. The paper does look much better now. I will raise my score.
>
> Also, minor points:
> *  the legend of Figure 2 seems to be broken. The colors of utility ranker and p(true), and those of semantic entropy and MSP are the same. Maybe the coral/orange color bar is for utility ranker, and the khaki color is for semantic entropy? You can try create a shared dictionary for each handle and their value for each subplot by `fig.legend(all_handles_labels.values(), all_handles_labels.keys())`.
> * please keep the updated pdf within 10 pages :).

---

> ### Author Response · Authors · 2024-11-28
>
> We thank you for your careful consideration of our responses, looking at the updated pdf, and the constructive feedback. Your feedback helped improve the quality and clarity of our work significantly. We have uploaded a pdf version with Figure 2's legend fixed and with content within 10 pages.

---

### Official Review · Reviewer_gKP3 · 2024-11-06

**Soundness:** 3
**Presentation:** 3
**Contribution:** 2
**Rating:** 6
**Confidence:** 3

**Summary:**

The paper proposes a straightforward approach of using a small passage utility model to improve the calibration of larger LLM-based QA models; i.e. it proposes a method to predict the reliability of the LLM answer based on the utility of the retrieved passages.

For a question $q$, the set of retrieved passages $R$ = $[p_1, p_2, ..., p_{|R|}]$, and a QA model $M$, the utility of a passage $p \in R$ is given by: $$ u = (a + e) / 2$$ where $a$ is the accuracy of the $M$ in predicting the ground-truth answer given passage $p$ and $e$ is the NLI entailment score of the question and predicted answer given the passage. A distillRoBERTa-based LM is trained to fit the utility scores. At inference time, the utility predictor assigns a score to each retrieved passage (given the question). The maximum utility score over all passages is used as the heuristic to abstain from answering.

The quality of different calibration techniques is compared on 4 QA datasets: NaturalQuestions, TriviaQA, WebQuestions, and SQuAD. The calibration techniques are compared on area under the rejection accuracy curve (calibration of abstaining) and AUROC of detecting incorrect answers. For two QA models, the trained utility predictor matches or improves over the performance of simple answer entropy-based heuristics. It is shown to be competitive with more complicated calibration techniques that rely on resampling multiple answers from the QA model.

An experiment is conducted to show that the utility predictor trained on NQ can be generalized out of distribution to SQuAD, PopQA, and RefuNQ. Moreover, the utility predictor can be used to rerank documents and improve QA accuracy.

**Strengths:**

1. The contribution is intuitive and straightforward. The utility predictor is shown to be a low-cost mechanism for improving the calibration of larger, more expensive QA models.
2. The chosen experiments are appropriate. The experiment testing generalization across datasets is valuable and adds to the strength of the approach.
    - Some concerns about "completeness" of experiments are raised int he next section
3. The connection of utility prediction to passage reranking should be studied further. This is especially important since the utility predictor is shown to be an effect reranker.
    - Can we use utility score as a stronger signal for rankers in general (not just for calibration)?

**Weaknesses:**

1. Several small details are missing in parts of the paper. Detailed questions are in the next section.
2. One big issue with the experiment set-up is that the QA models are not instructed to abstain. Thus, it is unclear how any of the calibration methods improve over the inherent ability of the QA models to abstain.
    - Under the current setup, even if the QA model abstains, it would be treated as "incorrect".
3. Please include a discussion of connections to "Evaluating Retrieval Quality in Retrieval-Augmented Generation" (SIGIR 2024)
    - They utilize a similar (query, passage) utility score for ranking retrieval systems

**Questions:**

1. What is the range of the utility scores? Based on the definition in Line 162, the value should be between [0, 1]. If so, then:
    1. Why do you need a sigmoid in Eq (2)?
    2. How are predicted utility scores $< 0$ or $> 1$ in Figure 1?
2. Sec 3.1: Is $y_M$ the predicted model answer given just passage $p$ or given the full set $R$?
3. Eq 1: Shouldn't the equation for margin loss be $max(0, m -y(u_i - u_j))$? i.e. the margin does not depend on $y$. What is actually implemented?
4. Line 181: It is unclear how important the BCE loss is. Please report the results of ablating the BCE loss.
5. Line 183: Is distillRoBERTa used as the utility predictor? How do you predict the utility score from the model? Please clarify the language in this line.
6. Line 200: Notation of utility predictor $v$ is misleading since $v$ does not depend on the model $M$ after it is trained. If I am misunderstanding this, please clarify.
7. Please include the ROC and RAC curves in the Appendix for completeness.
8. Line 259: How is the manual inspection performed? Over how many samples?
9. Line 269: It is unclear what you mean by "levels" here. Moreover, since all datasets are short-form QA, why do you believe clustering is affected?
10. Table 4: Please report OOD evaluation results of Utility Ranker (NQ) on TriviaQA and WebQuestions. The reported results of SQuAD are important, but it seems to be a setting where the utility ranker performed significantly better than all baselines. The distinction between different calibration approaches on the two other datasets is less clear.

Typos
---
- Line 134: Repeated "between"

---

> ### Author Response · Authors · 2024-11-20
>
> ## Weaknesses
>
> - We clarify questions below and will upload a new pdf version of the paper.
>
> - Inherent ability of the QA models to abstain.
>
> We did not explicitly instruct the QA models to abstain. It has been shown in previous work that LLM-based QA models instructed to abstain struggle with decisions on when they should or not refrain from answering [1]. That is, often abstain from answering when they should have provided an answer and generate a response when they should have abstained. Thus, to simplify the assessment of answer correctness, we did not instruct the models to abstain. In addition, following previous work [2], we treated the few observed abstentions as cases of answer uncertainty. We provide the percentage of abstentions out of the total incorrect ones for each model and dataset for completeness.
>
> [1] Examining LLMs' Uncertainty Expression Towards Questions Outside Parametric Knowledge
> https://arxiv.org/abs/2311.09731
> [2] Detecting hallucinations in large language models using semantic entropy
> https://www.nature.com/articles/s41586-024-07421-0#MOESM1
>
> - Connection to IR evaluation.
>
> We will include the discussion of the "Evaluating Retrieval Quality in Retrieval-Augmented Generation" (SIGIR 2024) paper.
>
> ## Questions
>
> - 1) Range of utility scores.
>
> Yes, utility scores based on entailment and accuracy are in the interval [0, 1]. These values are used to rank passages (best --higher-- to worse --lower--). However, although the ranking loss enforces the order constrain, it does not guarantees that predicted utility values will be in this interval. That is why predicted utility scores are $<0$ and $>1$ in Figure 1. Additionally, to enforce the signal on accuracy we add the auxiliary binary cross-entropy objective.
>
> - 2) To train the Utility Ranker we use the predicted model answer given just the passage $p$ because predictions are at passage level.
>
> - 3) Equation 3, margin loss. There is a typo, parenthesis are around i and js. The pytorch function used is MarginRankingLoss.
>
> - 4) We report ablation experiments on variations of the Utility Ranker training objective in the following Table. The different variants are trained with Gemma2-9B-Instruct as the target QA model on Natural Questions.
>
> |          |AUROC|AURAC|
> |----------|----------|----------|
> | $\mathcal{L}_{rank}, \text{utility is} (e + a)/2 + L_B $ | 0.77 | 0.76 |
> | $\mathcal{L}_{rank}, \text{utility is} (e + a)/2 $        | 0.67 | 0.70 |
> | $\mathcal{L}_{rank}, \text{utility is} (a) $              | 0.62 | 0.67 |
> | $\mathcal{L}_{rank}, \text{utility is} (e) $              | 0.67 | 0.70 |
> | $L_B$                   | 0.76 | 0.74 |
>
> where $e$ and $a$ are the entailment and accuracy scores, and $L_B$ is $\mathcal{L}_{BCE}$.
>
>
> - 5) The pre-trained model is used to initialise the encoder of the Utility Ranker. Thus, the Utility Ranker architecture consists of a BERT-based encoder with a pooling layer and two MLP layers stacked on top of the BERT-based encoder outputs. The output layer computes a utility score.
>
> - 6) Your understanding is correct. The idea was to reflect that $\upsilon$ was trained on data from the QA model $M$. As the notation is indeed misleading, we'll change to $\upsilon_M$.
>
> - 7) We will include these curves in the Appendix.
>
> - 8) We inspected a sample of size 50 for our task, in additional experiment we also used another critic model which agreed almost perfectly with Qwen2-70B used throughout our experiments. [1] originally inspected a sample of size 840.
>
> [1] Head-to-tail: How knowledgeable are large language models (LLMs)?
> https://aclanthology.org/2024.naacl-long.18/
>
> - 9) We mean different levels of detail, i.e., different amount of information. In these cases, the correct answers will not be clustered (despite being correct) what leads to falsely observing variation.
>
> - 10) We report OOD evaluation for all combinations of training and evaluation data (See Additional Results in response to Reviewer trrB).

---

> > ### Comment · Reviewer_gKP3 · 2024-11-24
> > **Reply to rebuttal**
> >
> > Thank you for clarifying my questions. Thank you for the new ablation results. They provide more support for the utility ranker objective. I will retain my original score for the following reasons.
> >
> > **RE: Allowing QA models to abstain:** I am not convinced by this argument. I believe that it is necessary to compare against the ability of the QA models to abstain. This is a baseline on the internal capability of the LM (without external control/mechanisms). If LMs are bad at abstaining, then it is all the more reason (beneficial to your argument) to include this baseline.
> >
> > **RE: Out-of-distribution generalization:** The new results point out that the OOD generalization of the utility ranker is quite limited. For example, in half of the settings, the performance drops to the level of using just the generation probability (MSP).

---

> ### Author Response · Authors · 2024-11-22
> **Response submitted**
>
> Dear Reviewer, thank you for your useful comments and recommendations. We have submitted a response addressing all your points and would like to have your acknowledgement/feedback and continue discussions. Kind regards, Authors.

---

### Official Review · Reviewer_trrB · 2024-11-07

**Soundness:** 2
**Presentation:** 2
**Contribution:** 2
**Rating:** 3
**Confidence:** 4

**Summary:**

This work looks at the task of uncertainty estimation in retrieval-augmented, open-domain question answering. The frame their task as predicting confidence estimates of a base retrieval-based QA model's predictions (experiments with Llama + Gemma) based on the set of retrieved passages and the input query. Their proposed approach is based on approach is based on training a small, separate *utility prediction* model that estimates the confidence in a base QA model prediction based on the input query and a single retrieved passage. To estimate confidence in a final prediction using a set of retrieved passages, they take the max predicted utility over all passages as the final confidence estimate.

To train this *utility prediction* model, the authors average (? -- see question below) the binary correctness score of the base QA model's prediction on a given question + retrieved passage and the predicted probability of question and QA model's predicted answer being entailed by the retrieved passage, treating this as a gold "utility value". The authors then train their smaller *utility prediction* to predict these utility values by summing two losses: (1) the BCE loss of the predicted utility against the gold utility and (2) a ranking loss between passage rankings obtained from the gold and predicted utility values.

In their experiments, the authors compare against calibration baselines (all are based primarily on using only the base LLM with sampling, prompting, and analyzing its predicted distributions). They train and evaluate on a variety of QA datasets using Gemma2, and see minor gains/losses when evaluating on NQ, TriviaQA, and Webquestions and a significant improvement when evaluating on SQUAD. The authors also repeat these experiments using LLAMA3 as the base QA model, and observe more mixed gains/losses over the baselines.

**Strengths:**

This work presents a method for uncertainty estimation in retrieval-based QA. Their method trains a separate smaller LM to estimate uncertainty in the base QA system's predictions based on a passage, question, and predicted answer. This system is trained on

**Weaknesses:**

## Related Work + Baselines
Similar methods that use small, additional trained models to estimate uncertainty have been proposed by [1] and [2] ([1]  is referenced in related work, but not compared against). Additionally, [3] has also noted the overlap between this passage utility / calibration task and similarly uses pretrained NLI models to verify / estimate uncertainty in QA system predictions. Given the similarity of these methods, they are important points of comparison to understand how this method differs and how it affects performance. See point below.

[1] Selective question answering under domain shift
Amita Kamath, Robin Jia, Percy Liang

[2] Knowing More About Questions Can Help: Improving Calibration in Question Answering
Shujian Zhang, Chenyue Gong, Eunsol Choi

[3] Can NLI Models Verify QA Systems' Predictions?
Jifan Chen, Eunsol Choi, Greg Durrett

## Evaluating role of the ranking loss and entailment score
A significant novelty from this work from the related works above is the usage of (1) an additional passage ranking loss (in addition to standard BCE loss) and (2) using entailment score in addition to answer correctness as a gold label to train the "passage utility predictor"; however, the role and usefulness of these changes are unclear. Additional ablation experiments would be helpful for understanding the impact of these changes and their benefits.

## Poor generalization to LLAMA3
While the results on Gemma2 seem promising, results using LLAMA3 as the base QA system are generally mixed/negative. Experimenting with more base QA systems and performing significance testing may help bolster these results.

**Questions:**

(Note in Summary) In L162, is this in-line equation supposed to be the average of accuracy and entailment score?

Why are generalization experiments were limited to only GEMMA and training on NQ and evaluating on SQuAD, PopQA, RefuNQ? It would be interesting to see the performance using LLAMA (especially givent he negative results here) and training + evaluation on a greater number of dataset combinations.

---

> ### Author Response · Authors · 2024-11-20
> **RE: Official Review of Submission11054 by Reviewer trrB**
>
> ## Weaknesses
>
> - Related Work + Baselines
>
> Thank you for these related work references. We will include a thorough discussion in our paper. Below we comment on them.
>
> A common observation on approaches [1, 2, 3] is that none of them is applied to retrieval augmented QA; but instead to Reading Comprehension (RC), i.e., the task of  generating an answer based on a positive (i.e., supposed to contain the answer) context document. Moreover, these approaches look at prediction at a single passage. In our work, we focus on the generalisation of passage utility to predict answer uncertainty (error) for retrieval augmented QA with a set of input passages.
>
> In relation to [1] and [2]. Their calibrator is trained to predict answer correctness (i.e., a binary classifier) from a context document based on shallow features (e.g., document length) plus QA model's output probabilities [1] or embeddings [2]. They assess the calibrator on distribution shift cases. In their scenario, all input documents are useful. In our scenario, the utility of retrieved passages is varied. Our calibrator will learn diverse causes of uncertainty. We show performance on in-distribution as well as OOD and adversarial QA settings. Interestingly, [1] observes that their approach does not capture unansarable questions while ours provides the best performance in these cases.
>
> In relation to [3]. This work relies on NLI models (off-the-shelf and QA-fine-tuned) to evaluate correctness of QA models' generated answers. The NLI usage and evaluation method in their work differs from ours. They focus on evaluating RC, i.e., after generating the answer, NLI is used to verify that the answer follows from the document. Instead, we use NLI as a metric to rank retrieved (potentially imperfect) passages. Furthermore, we train a secondary model to predict passage utility given a passage and user question (without the QA model generating an answer).
>
> Note, that there are two situations in our retrieval augmented LLM-based QA task in which only NLI verification is not enough. The quality of retrieved passages is not guaranteed. If the retrieved passage is related but misleading (e.g., contains a confounder entity), the answer produced by the QA LLM model can be entailed by the retrieved passage yet do not be the correct one. Second, given the amount of memorised knowledge in LLMs, there are cases where even the input passage does not entail the answer but the generated answer is still correct (i.e., the passage does not contain the answer but positively primes the model to generate the correct one). In our ablation experiments (see response to Reviewer gKP3), we show that using only entailment as a passage utility indicator in retrieval augmented QA helps but is not enough.
>
>
> [1] Selective question answering under domain shift Amita Kamath, Robin Jia, Percy Liang
> https://aclanthology.org/2020.acl-main.503.pdf
> [2] Knowing More About Questions Can Help: Improving Calibration in Question Answering Shujian Zhang, Chenyue Gong, Eunsol Choi
> https://aclanthology.org/2021.findings-acl.172.pdf
> [3] Can NLI Models Verify QA Systems' Predictions? Jifan Chen, Eunsol Choi, Greg Durrett
> https://aclanthology.org/2021.findings-emnlp.324.pdf
>
>
> - Evaluating role of the ranking loss and entailment score
>
> The significant novelty of our work lies in the prediction of answer uncertainty (error) for retrieval augmented QA with $|R|$ input passages and that we do this from individual passage utilities. In the ablation experiments (see response to Reviewer gKP3), we show the impact of the ranking loss and the usage of NLI scores for the ranking signal.
>
>
> - Poor generalization to LLAMA3
>
> We report results with an additional base LLM of similar size but different family, i.e., Mistral-7B-Instruct-v0.3. Development results in the 'Additional Results' post.
>
> We will add results on the same family but different sizes for Gemma2 in the new version of the pdf (we are currently running these experiments).
>
> ## Questions
>
> - Equation in L162 is the average of the accuracy and entailment score.
>
> - Generalisation experiments with more combinations of train/test data and models.
>
> We report additional experiments with all combinations of train/test data for experiments on distribution shift in the Table in the 'Additional Results' post.
>
> We will add results for Llama-3.1 on the adversarial QA datasets (PopQA and RefuNQ) in the new version of the pdf.

---

> ### Author Response · Authors · 2024-11-20
> **Additional Results**
>
> - Additional base LLM, Mistral-7B-Instruct-v0.3.
>
> |        |NaturalQuestions|        |TriviaQA|        |WebQuestions|         |SQuAD|        |
> |--------|--------|--------|--------|--------|--------|--------|--------|--------|
> |        |AUROC|AURAC|AUROC|AURAC|AUROC|AURAC|AUROC|AURAC|
> | PPL | 0.65 | 0.69 | 0.65 | 0.80 | 0.62 | 0.70 | 0.66 | 0.65  |
> | MSP | 0.70 | 0.71 | 0.74 | 0.82 | 0.67 | 0.73 | 0.72 | 0.68  |
> | PMI | 0.49 | 0.60 | 0.57 | 0.76 | 0.56 | 0.68 | 0.54 | 0.58  |
> | p(true) | 0.73 | 0.71 | **0.80** | **0.85** | 0.69 | 0.75 | 0.70 | 0.67  |
> | Regular Entropy | 0.65 | 0.69 | 0.66 | 0.80 | 0.63 | 0.71 | 0.70 | 0.68  |
> | Cluster Assignment | 0.71 | 0.72 | 0.76 | 0.82 | 0.71 | 0.75 | 0.75 | 0.69  |
> | Semantic Entropy | 0.72 | 0.72 | 0.77 | 0.83 | 0.71 | 0.74 | 0.75 | 0.70  |
> | Ans.Len  | 0.65 | 0.68 | 0.69 | 0.80 | 0.64 | 0.72 | 0.66 | 0.64  |
> | Retriever Score | 0.59 | 0.65 | 0.61 | 0.77 | 0.58 | 0.69 | 0.64 | 0.63  |
> | Utility Ranker | **0.76** | **0.74** | 0.77 | 0.84 | **0.73** | **0.77** | **0.80** | **0.72**  |
>
>
> - Additional experiments on distribution shift.
>
> |      | NaturalQuestions |  |TriviaQA |  |WebQuestions |   | SQuAD | |
> |------|------|------|------|------|------|------|------|------|
> |      | AUROC | AURAC |AUROC | AURAC |AUROC | AURAC |AUROC | AURAC |
> | NaturalQuestions | **0.76** | **0.72** | 0.72 | 0.86 | 0.65 | 0.67 | 0.72 | 0.68 |
> | TriviaQA | 0.64 | 0.67 | **0.81** | **0.88** | 0.63 | 0.68 | 0.71 | 0.68 |
> | WebQuestions | 0.60 | 0.64 | 0.72 | 0.86 | **0.72** | **0.71** | 0.58 | 0.59 |
> | SQuAD | 0.65 | 0.67 | 0.77 | 0.87 | 0.61 | 0.65 | **0.81** | **0.74** |
>
> The first column indicates the train data, the first row indicates the evaluation data. Results in the diagonal correspond to the Utility Ranker trained/evaluated in the same data.

---

> ### Author Response · Authors · 2024-11-22
> **Response Submitted**
>
> Dear Reviewer, thank you for your useful comments and recommendations. We have submitted a response addressing all your points and would like to have your acknowledgement/feedback and continue discussions. Kind regards, Authors.

---

> ### Author Response · Authors · 2024-11-28
>
> Dear Reviewer,
>
> We have uploaded a new pdf version of our paper with further details (with major changes as outlined in our general comment).
>
> As the extended discussion phase ends in three days, we kindly request confirmation of receipt of our response and updated pdf and welcome any additional feedback.
>
> Sincerely, The Authors

---

### Official Review · Reviewer_NaAL · 2024-11-08

**Soundness:** 3
**Presentation:** 3
**Contribution:** 2
**Rating:** 5
**Confidence:** 4

**Summary:**

The paper proposes a novel approach for answer error prediction in retrieval augmented question answering. The premise is that the the retrieved passages and their interaction with the QA model’s parametric knowledge is a strong indicator of answer correctness. To measure this as a utility score for each passage, a small neural network is trained using a ranking loss - where the maximum utility score for each passage is the estimate for answer error prediction. On a few existing QA benchmarks (Natural Questions, TriviaQA, WebQuestions, SQuAD), this is shown to to be better than existing error prediction approaches based on entropy and resampling, while being more compute efficient.

**Strengths:**

- The approach to train a separate smaller neural network to predict passage utility scores is novel. The construction of the data and loss for the scoring model, using entailment and accuracy is also intersting and original.
- The paper provides an efficient way to predict the error rate at an example level, which could be very useful for latency sensitive systems in order to make a triggering decision for question answering.
- The overall flow of the paper is good, it is succinctly written, and the experimental results are compelling and clearly presented.
- The paper also touches upon the reranking approach to improve the performance of QA model using their utility scoring model, which seems potentially useful for some applications.

**Weaknesses:**

- One strong shortcoming of this approach is where multiple passages are needed to correctly answer the question, i.e. using multihop reasoning. In such cases, the utility both each of the passages in isolation could be low, and hurt the error prediction. Most of the baselines that use the entire passage set would be robust to this.
- The modeling utility scores used to create  the ranking dataset has room for improvement. The scores could have smoother accuracy or entailment values instead of the binary values. And other, more principled aggregation functions could be explored instead of a simple average.
- The evaluation for the utility ranker seems weak. The baseline in table 5 is not reranking at all. A better baseline could be a different utility ranker trained using the neural network, possibly with a simple objective such as predicting the error rate of the neutral network given x and p.

**Questions:**

- In line 200, why is e arg max? Could be a typo.
- Did you compare the inference time difference between your approach and the baseline? It would be useful to see that comparison as well, since that's one of the key claims made.

---

> ### Author Response · Authors · 2024-11-20
>
> ## Weaknesses
>
> - Utility Ranker to predict on multi-hop reasoning.
>
> In our paper, we focused on short-form QA and single hop reasoning. However, as you correctly point out in multi-hop QA, no single passage will have high utility. In this task setting, we expect that passage utilities will be middle/low for relevant passages and much lower for irrelevant ones. Thus, several points follow from this. First, this case highlights the need to train the Utility Ranker with a smoother score like NLI. Second, while we use a simple passage utility aggregation function to predict answer uncertainty (error) for retrieval augmented QA, passage utilities could be used as features of an answer uncertainty predictor. Third, our Utility Ranker could be useful for rearranging input passages to improve QA performance in multi-hop QA. We will run further experiment on HotPotQA [1] and include results in the final version of the paper.
>
> [1] HotpotQA: A Dataset for Diverse, Explainable Multi-hop Question Answering
> https://aclanthology.org/D18-1259/
>
> - Reference utility score to train the Utility Ranker.
>
> We would like to clarify that we rely on smooth values as we include the entailment score, i.e., the posterior probability of the entailment class. As for the aggregation functions, we focus on a simple function as already this shows that it is possible to obtain comparable performance to other more expensive uncertainty estimation methods. However, it would make sense to define more complex aggregation methods, e.g., training a confidence estimator based on the set of passage utilities together with other features like the Maximum Sequence Probability (MSP).
>
> - No other re-ranking baseline.
>
> The main evaluation of the Utility Ranker is extensive covering in-distribution, our-of-distribution, and adversarial QA in various datasets and models as well as ablation studies. The main focus of our work, is on predicting answer uncertainty (error). Thus, we evaluate this w.r.t. to strong baselines. As an additional experiment, we show that the Utility Ranker also brings added value to improve QA performance on top of the original ranking provided by the Information Retrieval (IR) system. Thus, we do not include baselines/comparison systems in this additional experiment. Nevertheless, we could include in the final version experiments with other re-ranking baselines for completeness.
>
> ## Questions
>
> - L200 argmax.
>
> Yes, there is a typo, it should be max(.), i.e., take the maximum predicted utility.
>
> - Inference time difference between approaches.
>
> There is indeed a difference in latency at inference time. Below we detail the number of forward passes (and what type of forward call) required by each approach.
>
> |Approach | Nb. of inference passes|
> |-----------|-----------|
> | PPL | 1 LLM-G |
> | MSP  | 1 LLM-G |
> | PMI  | 2 LLM-G |
> | p(true) | (N + 1) LLM-G + 1 LLM-E |
> | Regular Entropy   | (N + 1) LLM-G |
> | Cluster Assignment   | (N + 1) LLM-G + 1 LLM-E |
> | Semantic Entropy   | (N + 1) LLM-G + 1 LLM-E |
> | Ans.Len  | 1 LLM-G |
> | Retriever Score | 0 LLM-G   |
> | Utility Ranker | size-of(R) Bert-F |
>
> Where LLM-G means answer generation with a QA prompt ( $|R|$ passages and question), LLM-E means evaluation with a verification prompt (including as many in-context examples as possible, question, and candidate answers). Bert-F means an Utility Ranker forward on passage and question to obtain the passage utility score.  size-of(R) means $|R|$.

---

> ### Author Response · Authors · 2024-11-22
> **Response Submitted**
>
> Dear Reviewer, thank you for your useful comments and recommendations. We have submitted a response addressing all your points and would like to have your acknowledgement/feedback and continue discussions. Kind regards, Authors.

---

> ### Author Response · Authors · 2024-11-28
>
> Dear Reviewer,
>
> We have uploaded a new pdf version of our paper with further details (with major changes as outlined in our general comment).
>
> As the extended discussion phase ends in three days, we kindly request confirmation of receipt of our response and updated pdf and welcome any additional feedback.
>
> Sincerely, The Authors

---

### Official Review · Reviewer_zmTA · 2024-11-08

**Soundness:** 3
**Presentation:** 2
**Contribution:** 2
**Rating:** 5
**Confidence:** 3

**Summary:**

This paper proposes an approach to measure the uncertainty in Retrieval Augmented Question Answering tasks.
Concretely, they train a small neural network called utility ranker which assigns a score for each retrieved passage from a given retriever to judge if the retrieved passage is useful for the answer generated by some QA model.
The authors show that this approach is on par or better than existing error prediction approaches while being light-weight at the same time.

**Strengths:**

The authors ran experiments with 2 QA models and for a lot of the settings the utility ranker outperforms existing metrics in terms of uncertainty estimations of the retrieved passages.
Experiments results also suggest that the method they proposed is also robust to OOD datasets where the ranker is not trained on.

**Weaknesses:**

- Many of the notations are unclear. See in Questions.
- QA models used for evaluation only limit to Gemma2-9b-instruct and Llama3.1-8b-instruct which are of similar size. More experiments should be done using models with various sizes to see if similar conclusions still hold.
- For Llama3.1-8b-instruct, results from table3 seems to suggest that Utility Ranker is not doing better than just looking at the probability of generating the next token to be "True". Is the training of this ranker really necessary?

**Questions:**

1. in (1), is m some hyper-parameter introduced in the model? If it was taken from other works, where did it come from? If it is optimized for this task, how did you optimize?
2. the i and js from L_{rank} are never summed up in the total loss term. But I assume you do this for each retrieved passage pair, is that the case?
3. How is the accuracy a defined at the bottom of page 3?

---

> ### Author Response · Authors · 2024-11-20
>
> ## Weaknesses
>
> - We clarify notations below and will upload a new pdf version of the paper.
>
> - Experiments on various sizes.
>
> We add an additional model of a different family but similar size, i.e., Misral-7B-Instruct-v0.3 (See response to Reviewer trrB).
> We will add variants of different sizes, i.e., 2B and 27B, for Gemma2.
>
> - Llama3.1-8b-instruct, Utility Ranker on par with p(True) approach (Table 3).
>
> Our approach is an alternative that performs comparable (and sometimes even better) to strong but more expensive methods. P(True) is not a simple probability of True approach. To work well (as reported in our paper), it requires as many in-context examples as possible and a question that is formulated based on ten samples (as proposed by [1]). Thus, it requires (i) generating ten samples with a potentially big LLM and a long retrieval augmented context plus (ii) the final forward with a huge prompt (in-context examples and the actual question). See cost of each approach in response to Reviewer NaAL.
>
> [1] Language models (mostly) know what they know
> https://arxiv.org/abs/2207.05221
>
> ## Questions
>
> - $m$ in Equation 1 is a hyper-parameter set to 0.1 in all our experiments. In initial experiments we search with values 0.01/0.001 but results were not better.
>
> - Yes, i and js in the equation are for each pair. In the new pdf version we rewrite the equation for clarity.
>
> - The accuracy $a$ is the observed accuracy of the target QA model on input question $x$ and passage $p$, ($x$, $p$). To train the Utility Ranker we generate training data with the target QA model.

---

> > ### Comment · Reviewer_zmTA · 2024-11-26
> >
> > Thanks for the reply from the authors. Some of my questions and concerns are addressed.
> > RE: additional results on Mistral-7B-Instruct-v0.3:
> > I agree that Utility Ranker is a relatively light-weight method that is on par with the P(True) method which is a much more expensive method.
> >
> > RE: notations
> > thanks for the explanation and please include more details in the revised version.

---

> ### Author Response · Authors · 2024-11-22
> **Response Submitted**
>
> Dear Reviewer, thank you for your useful comments and recommendations. We have submitted a response addressing all your points and would like to have your acknowledgement/feedback and continue discussions. Kind regards, Authors.

---

> ### Author Response · Authors · 2024-11-28
>
> Thank you for considering our response. We have uploaded a new pdf version of our paper with further details (with major changes as outlined in our general comment). We kindly ask you to let us know if your concerns have been addressed.

---

### Author Response · Authors · 2024-11-28

Dear Reviewers,

We would like to thank you all for your thoughtful review process and valuable comments that helped to improve the quality and clarity of our paper. We would like to summarise the major discussion points that have been addressed in the re-submitted pdf.

- **Clarification of the contributions of our work.** We have updated the last two paragraphs of the introduction to make clearer the contributions of this work. We have incorporated in Appendix C.1 and Table 8 a description on the execution cost of each comparison uncertainty estimation method.

- **Additional target QA model** We have incorporated uncertainty quantification evaluation for and additional target QA model, namely *Mistral-7B-Instruct-v0.3* (Section 5.1 Table 1).

- **Ablation study on the Utility Ranker training objective** We have incorporated an ablation study on the different components of the training objective (Appendix D.1 Table 9) for the three target QA models.

- **Complete results under distribution-shift evaluation settings** We incorporated a zero-shot assessment of the Utility Ranker on different combinations of training/evaluation data (Section 5.2) and extended the results discussion about the adversarial QA tasks.

- **Complete experiments on improving QA performance** We have added results for additional datasets and another baseline.

- **General presentation** We incorporated more details on the description of the our approach (Section 3.1 and 3.2). We have improved the presentation of the results in Section 5.1 (added Figure 2). We incorporated a discussion of the suggested related work (L153 and L251).

We kindly request confirmation of receipt of our responses and the updated pdf and welcome any additional feedback.

Thank you for your time and consideration.

Sincerely,
The Authors

---

### Meta-Review · Area_Chair_dnRy · 2024-12-21

**Metareview:**

This paper introduces a new uncertainty quantification method for retrieval-augmented question answering. This method trains a small neural model to explicitly compute the utility of individual input passages for a downstream QA model. Experimental results on six QA datasets demonstrate that this method achieves performance comparable to existing sampling-based uncertainty quantification methods, while being significantly more efficient at test time.

Strengths
- The paper tackles the important problem of quantifying uncertainty at the example level (NaAL, gKP3, 575d).
- The paper introduces a new approach by training a smaller neural model to directly evaluate passage utility (NaAL, 575d).
- Experiments are comprehensive, including diverse settings and demonstrating generalization to OOD setups (zmTA, gKP3, 575d).

Weaknesses
- Insufficient discussion / comparison to prior work, including selective question answering, calibration, and reranking in QA, which are virtually the same problem (trrB, gKP3).
- No ablation studies on key components, such as the passage ranking loss and the use of entailment scores, which are key differentiators from previous work (trrB).
- Missing baselines, such as predicting error rates directly for a given question and passage (NaAL) or enabling the QA model to abstain from answering (gKP3).
- Difficult to generalize when multiple passages are jointly needed, e.g., multi-hop QA (NaAL).
- Poor generalization to LLMA3 (trrB).
- Unclear notations, lack of details, and unclear description of the method and experiments (zmTA, gKP3, 575d)

**Additional Comments On Reviewer Discussion:**

Author responses do not sufficiently resolve reviewers' concerns.

---

### Decision · Program_Chairs · 2025-01-22

Reject